# SAPIENT: Continual Test-time Adaptation via Lightweight plug-and-play Adapters

## Abstract

Continual test-time adaptation (TTA) is the problem of adapting a pre-trained source model at inference-time to handle test samples from a non-stationary distribution, while not forgetting the knowledge acquired from earlier domains. Existing continual TTA methods either make unsupervised test-time updates to the entire model, which can be expensive and prone to forgetting, or do so by keeping the base model frozen and adding a small number of learnable adapter modules for better time/memory efficiency and mitigating forgetting. We present SAPIENT (continual teSt-time adaPtation vIa lightwEight plug-aNd-play adapTers), a parameter-efficient adapter based approach which not only offers the usual benefits of the adapter based continual TTA methods, but offers additional key benefits, such as (1) its simple plug-and-play design seamlessly integrates with various continual TTA losses, making our approach complementary to existing continual TTA methods, improving their time/memory efficiency and knowledge retention, (2) it does not require access to the source domain data unlike recent adapter based continual TTA methods, and (3) its parameter-efficiency also makes it computationally feasible to design its Bayesian extensions which can help in estimating the uncertainty in adapter weights, which in turn yields more robust predictions. Through extensive experiments on a segmentation task and four classification tasks for continual TTA, we demonstrate that, with substantially ($\sim$90%) fewer trainable parameters, our method achieves competitive or better performance compared to the evaluated continual TTA baselines, resulting in efficient and robust adaptation and inference at test-time.

## 1 Introduction

Real-world applications of deep learning models routinely encounter test data that may come from a non-stationary distribution different from the source training data distribution. For example, when deployed in the wild, a model trained on clean images may observe various domain shifts, such as low-light situations, camera flares, etc., at test time. In such settings, the source pre-trained model must adapt at test (inference) time without access to labeled data from the test domain. This problem setting is known as *test-time adaptation* (TTA) (Liang et al., 2023; Sun et al., 2020; Liang et al., 2020; Liu et al., 2021; Wang et al., 2021; Zhou and Levine, 2021; Prabhu Teja and Fleuret, 2021). Moreover, doing so in a setting when the test domain itself may continuously undergo a shift over time is even more challenging; in this setting, we need to ensure that the model performs well on the new domain(s) while also not suffering from forgetting on the previously seen domains in order to maintain its predictive accuracy on test inputs from previous domains. This problem setting is referred to as *continual* TTA and has received significant recent interest (Wang et al., 2021; Niu et al., 2022; Hong et al., 2023; Song et al., 2023).

Some of the prior works on TTA (Wang et al., 2021) have shown that updating only the batch norm layers works reasonably well. This is also desirable for efficient inference at test time. However, in the more challenging *continual* TTA setting, there may be error accumulation over time (Wang et al., 2021), which usually necessitates continual TTA methods to update all the model weights for each new test domain, such as CoTTA (Wang et al., 2022). Although such full model updates perform better than batch norm update methods, inference efficiency also suffers. To address this, recent works in continual TTA (Song et al., 2023) have explored adapter-based methods (Rebuffi et al., 2017; Rosenfeld and Tsotsos, 2018; Houlsby et al.,

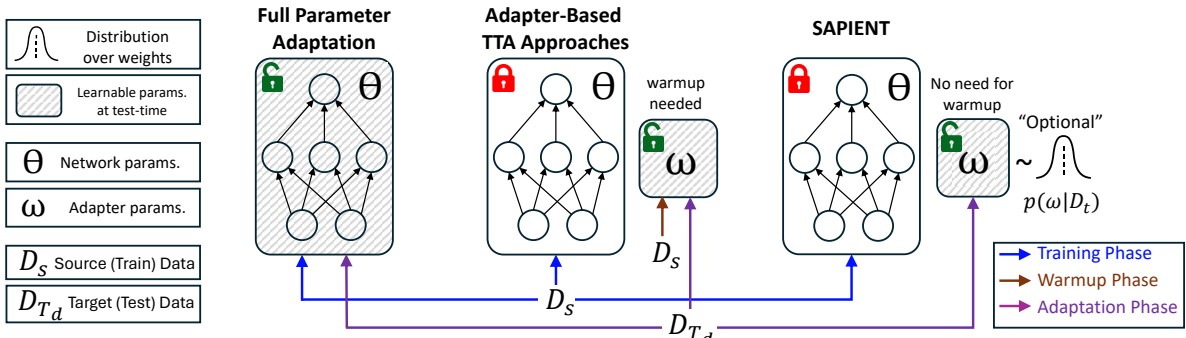

Figure 1: SAPIENT vs full parameter update based methods and other existing adapter update based methods for continual TTA. In SAPIENT, the designed adapters can be added to any existing pretrained model without needing a warmup phase for initializing adapter parameters, which eliminates dependency on source dataset. SAPIENT also allows estimating the uncertainty in adapter weights by learning their posterior distribution.

2019; Chen et al., 2022) where the source model weights are kept frozen, and a few adapter layers consisting of a small number of parameters, trained using an unsupervised loss, are inserted in the source model to handle the domain shifts. However, despite parameter-efficiency, the adapter-based models (Song et al., 2023) aren't really *plug-and-play* as they usually rely on the availability of the source data (Fig. 1) to learn a good initialization (warmup) of the adapter parameters in the adaptation phase. This also tends to make the new model drift towards the source during warmup phase. Moreover, the adapter based approaches can have a tendency to be overconfident as their finetuning is based on very small amounts of unlabeled test inputs, necessitating proper uncertainty quantification.

We present SAPIENT (continual teSt-time adaPtation vIa lightwEight plug-aNd-play adapTers), a framework that enables parameter-efficient continual TTA, addressing the aforementioned challenges that existing full-network update as well as adapter update based continual TTA methods suffer from. SAPIENT is driven by the goal of being both efficient and robust. Its adapter-based parameter-efficient design achieves time/memory efficiency (requiring updating only adapter parameters at inference time) and its ability for uncertainty quantification (via Bayesian extensions) provides it robustness. SAPIENT also uses an identity-equivalent initialization scheme, which avoids the need for access to the source domain data when updating the adapter parameters, making it a plug-and-play approach. Another appealing aspect of SAPIENT is that it seamlessly integrates with a variety of unsupervised TTA losses proposed in continual TTA literature, making existing continual TTA methods (Wang et al., 2021; Song et al., 2023; Niu et al., 2022) readily benefit from the advantages offered by SAPIENT and thus SAPIENT is complementary to such methods. We evaluate SAPIENT on various widely used benchmarks in continual TTA and show that the existing methods can be made parameter efficient by a massive margin (∼**90% fewer trainable parameters**) with comparable or better performance and much better knowledge retention (Fig. 2).

Our contributions are summarized below:

- We present SAPIENT, a new parameter and memory efficient framework for continual TTA. SAPIENT appeal lies in its plug-and-play approach which enables its easy integration with existing continual TTA methods to improve their efficiency and robustness.

- SAPIENT also benefits from an *identity-equivalent* initialization mechanism for the adapter parameters, which helps mitigate the dependency on the availability of the source dataset.

- We show that the parameter-efficiency of SAPIENT also helps designing its Bayesian extensions for estimating the uncertainty in adapter weights, which in turn yields more robust predictions.

- With extensive experiments on the widely used challenging segmentation benchmark (Cityscapes-to-ACDC) and classification benchmarks (CIFAR10C, CIFAR100C, ImageNetC, and ImageNet3DCC) for continual TTA, we show that the existing TTA approaches can be made more parameter efficient with no or meager drop in adaptation performance.

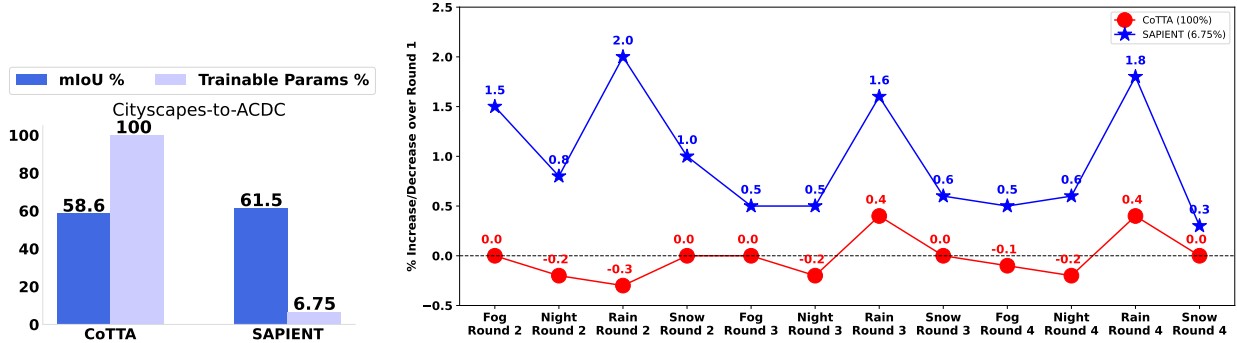

Figure 2: A comparison with continual TTA with full network update (CoTTA used as a baseline) vs adapter based update (SAPIENT). **Left:** mIoU (%) for Cityscapes-to-ACDC along with % of trainable parameters. Higher is better for mIoU, whereas lower is better for the trainable parameter percentage. **Right:** Knowledge-retention ability of CoTTA vs SAPIENT - the improvement over round 1 performance is shown for Cityscapes-to-ACDC when test domains are revisited (also see Table 1 in Experiments section).

## 2 Problem Setup and Formulation

Assuming a classification setting, let $\mathcal{D}_\mathcal{S} = \left\{ \mathbf{x}_s^{(m)}, y_s^{(m)} \right\}_{m=1}^M$ denote the source domain ($\mathcal{S}$) data used to train a model $f_\theta$ where $\theta$ denotes the model parameters. The model $f_\theta$ takes an input $\mathbf{x}$ and predicts the most likely label $\hat{y}$. We denote the source domain pre-trained model parameters as $\theta_s$. The model $f_\theta$ is pre-trained using source domain data as $\theta_s = \min_\theta \frac{1}{M} \sum_{m=1}^M \mathcal{L}_\mathcal{S} \left( f_\theta \left( \mathbf{x}_s^{(m)} \right), y_s^{(m)} \right)$, where $\mathcal{L}_\mathcal{S}$ is some suitable supervised loss function for source domain $\mathcal{S}$ data, and $y_s^{(m)}$ is the ground truth label for input $\mathbf{x}_s^{(m)}$. At test-time, we assume access to only $\theta_s$ and do not assume access to source domain data. Our goal is to adapt the source model $\theta_s$ to predict the labels of unlabeled test inputs from a different distribution.

Formally, let $\mathcal{D}_{\mathcal{T}_d} = \{ \mathbf{x}_d^{(n)} \}_{n=1}^{N_d}$ denote the test data from the target domain $\mathcal{T}_d$. To predict the labels of these test inputs, we update $\theta_s \to \theta_d$ by minimizing an *unsupervised* TTA loss

$$\theta_d = \min_\theta \frac{1}{N_d} \sum_{n=1}^{N_d} \mathcal{L}_{\mathcal{T}_d} \left( f_\theta \left( \mathbf{x}_d^{(n)} \right) \right), \tag{1}$$

where $\mathcal{L}_{\mathcal{T}_d}$ can be an appropriate unsupervised loss, such as the pseudo-label based entropy loss in TENT (Wang et al., 2021), weighted entropy loss in EATA (Niu et al., 2022), student-teacher cross-entropy loss in CoTTA (Song et al., 2023), and various other losses. Having obtained $\theta_d$, we make predictions for a test input $\mathbf{x}_d$ from the new domain as $\hat{y} = f_{\theta_d}(\mathbf{x}_d)$. In continual TTA (Wang et al., 2022; Song et al., 2023), the test examples can arrive sequentially from different target domains $\{\mathcal{T}_d\}_{d=1}^D$, making the distribution of test data non-stationary. Moreover, we assume that the learner gets no information about the switch in the domain. These aspects make continual TTA considerably more challenging than standard TTA. Further, in *online* continual TTA, the learner gets to see the test input only once, and multiple passes are not allowed.

To handle the above issues, and the error accumulation and catastrophic forgetting of the earlier domains caused by the continuously drifting test domain distributions, we present a framework based on augmenting a base network with a small number of trainable adapter parameters (Houlsby et al., 2019; Hu et al., 2021; Varshney et al., 2021). The base network is kept frozen at test time, and only the adapter parameters are updated using an unsupervised loss. Updating only the adapters significantly improves the parameter efficiency without compromising the performance. Moreover, because of its parameter-efficiency, we show that unlike other recent approaches to continual TTA, our framework can be easily extended to estimate the uncertainty in the adapter parameters by taking a Bayesian approach (which would be infeasible to do for the entire model (Yang et al., 2023; Onal et al., 2024)) and computing the posterior distribution over the adapter parameters. This yields more robust predictions by the adapted model.

# 3  A Plug-and-Play Approach to Continual TTA

In this section, we provide a detailed description of our framework SAPIENT. While our framework is general and can be applied to a variety of base architectures (we experiment with both Transformers and CNNs) for vision as well as NLP tasks, here we describe it assuming a feed-forward base architecture in which we employ groupwise and pointwise convolutional filters between layers (as shown in Fig. 1). These additional filters, which consist of only a small number of additional parameters, act as adapters and can be efficiently updated at test time given unlabeled input(s) from a new distribution by optimizing any suitable unsupervised TTA objective (Eq. 1).

## 3.1  SAPIENT: Continual Test-time Adaptation via Lightweight Plug-and-Play Adapters

Our approach SAPIENT introduces lightweight adapters to the base network and, at test time, only updates the adapter parameters using an unsupervised TTA loss. For the choice of a lightweight adapter, the primary objective is parameter efficiency, with comparable or better predictive accuracy. In our work, we specifically design adapters considering the requirement for making them compatible with identity transformation to remove the dependency on the availability of the source training dataset for initial warmups (see section 3.2 for more details). Primarily, given a pre-trained model (also referred to as *base model* in TTA setup) with parameters $\theta_s$, we insert new adapter parameters ($\omega$) in between layers.

Considering every layer in the base model ($\theta_s^{(l)}$) acting as a sequence of feature transformations over the input, we insert adapters ($\omega_s^{(l)}$) after the feature transformations. For instance, consider a sequence of transformations present in the base model:

$$\mathbf{F}^{(l-1)} \to \theta_s^{(l)} \to \mathbf{F}^{(l)} \to \dots \; \mathbf{F}^{(l+n-1)} \to \theta_s^{(l+n)} \to \mathbf{F}^{(l+n)}$$

where $\mathbf{F}^{(l)}$ represents the transformed features after the $l^{th}$ layer of the base model ($\theta_s^{(l)}$). Note $\mathbf{F}^{(l)} \in \mathbb{R}^{h \times w \times c}$ here denotes the feature map with $h$, $w$ and $c$ as its the height, width, and number of channels, respectively, where the parameters $\theta_s^{(l)}$ define a convolution operation $g_{\theta_s^{(l)}}(\mathbf{F}^{(l-1)})$. After inserting adapters in between, we obtain the sequence:

$$\mathbf{F}^{(l-1)} \to \theta_s^{(l)} \to \mathbf{F}^{(l)} \to \omega^{(l)} \to \mathbf{F}_\omega^{(l)} \to \dots \; \mathbf{F}^{(l+n-1)} \to \theta_s^{(l+n)} \to \mathbf{F}^{(l+n)}$$

where $\mathbf{F}_\omega^{(l)}$ depicts the transformation made by the newly added adapters. In practice, we only insert the adapters in a few locations, depending on the computational and memory budget. For the $l^{th}$ layer, we denote the frozen parameters and adapter parameters using $\theta_s^{(l)}$ and $\omega^{(l)}$, respectively. For brevity of notation, we omit $l$, and use $\theta_s$ and $\omega$ to denote $\theta_s^{(l)}$ and $\omega^{(l)}$, respectively. we propose using a combination of pointwise and groupwise convolution for adapter modules, which can be used as lightweight, parameter-efficient adapters. Groupwise COnvolution (GCO) using $r$ number of groups, where $r \ll c$ reduces the number of parameters by a considerable margin, requiring only $\frac{c}{r}$ times fewer parameters than the standard convolution filter. In contrast, PointWise Convolution (PWC) helps handle the drawback of GCO capturing fewer feature maps. For PWC, we use convolution filters of size $1 \times 1 \times c$, which are $9\times$ more parameter efficient than standard $3 \times 3$ convolution operation. Combining both operations (GCO and PWC) makes the transformation parameter efficient by a significant margin.

We use $g_{\omega_{\mathrm{G}}}$ to denote $3 \times 3$ GCO operation of group size $r$ having adapter parameters $\omega_{\mathrm{G}}$ and $g_{\omega_{\mathrm{P}}}$ to denote the PWC operation having adapter parameters $\omega_{\mathrm{P}}$. GCO and PWC modify the feature map as $\mathbf{F}_{\mathrm{G}}^{(l)} = g_{\omega_{\mathrm{G}}}\left(\mathbf{F}^{(l)}\right)$ and $\mathbf{F}_{\mathrm{P}}^{(l)} = g_{\omega_{\mathrm{P}}}\left(\mathbf{F}^{(l)}\right)$, respectively. Further, we use $\omega$ to collectively denote $\omega_{\mathrm{G}}$ and $\omega_{\mathrm{P}}$. A noteworthy point about the proposed mechanism is that the base model architecture needs no modifications for insertion of these adapters since the dimensions of $\mathbf{F}_{\mathrm{G}}^{(l)}$ and $\mathbf{F}_{\mathrm{P}}^{(l)}$ are the same as the incoming feature map $\mathbf{F}^{(l)}$. Combining $\mathbf{F}_{\mathrm{G}}^{(l)}$ and $\mathbf{F}_{\mathrm{P}}^{(l)}$, we obtain the final transformed feature map, $\mathbf{F}_{\mathrm{A}}^{(l)}$, using adapters as $\mathbf{F}_{\mathrm{A}}^{(l)} = \mathbf{F}_{\mathrm{G}}^{(l)} \oplus \mathbf{F}_{\mathrm{P}}^{(l)}$, where $\oplus$ denotes element-wise addition (also see Fig. 3 (Left), for a visual representation of the proposed operation).

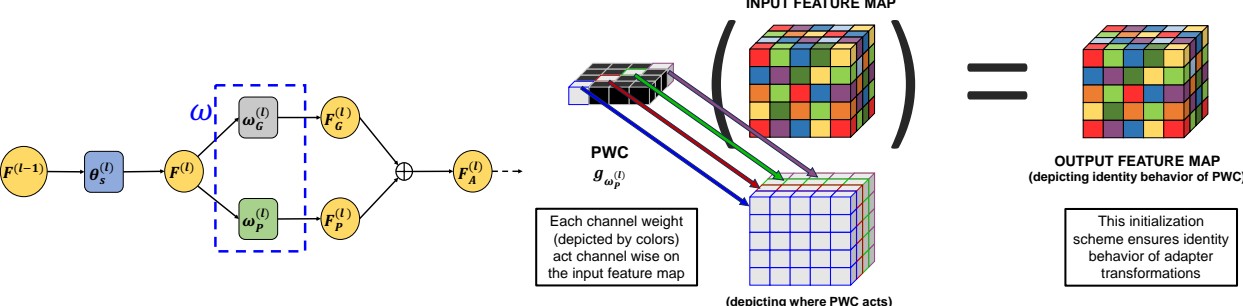

Figure 3: **Left.** Groupwise COnvolution (GCO) and a PointWise Convolution (PWC) are inserted between frozen backbone layers. Their outputs are fused via element-wise addition, $F_A^{(l)} = F_G^{(l)} + F_P^{(l)}$, to produce the adapted feature map. Only these adapter parameters are updated at test time, enabling efficient adaptation to non-stationary target distributions without modifying the source backbone or requiring source data. **Right.** The figure highlights the proposed initialization scheme for the added adapter parameters. Specifically, the Point Wise Convolution (PWC), when initialized with diagonal elements (highlighted in white), acts channelwise on the input feature map, making the interaction between the channels zero and projecting the same feature space to act as the output feature map.

## 3.2 Preserving Source Knowledge with Identity Equivalent Initialization

Introducing adapters after any layer of the pre-trained source model can affect the feature representations of source data. Therefore, an initialization scheme is required for the adapter parameters to ensure that the source feature representations are not affected. One way to address this issue is to initialize the adapter parameters using a warmup training done on the source dataset (Song et al., 2023). However, this requires access to the source dataset. In SAPIENT, we address this issue by introducing a new initialization strategy which avoids the need of access to the source data.

Specifically, we make the initial configuration of the adapter parameters equivalent to an *identity function*. The parameters of GWC ($\omega_G$) are initialized with zeros. For PWC parameters ($\omega_P$), we initialize it with a four-dimensional tensor $\mathcal{Q}$ (input channel $\times$ output channel $\times$ kernel size $\times$ kernel size) where kernel size is 1, and the first two dimensions reflect an identity matrix for no interaction between the channels. Thus, initially, we have $\mathbf{F}_G^{(l)} = g_{\omega_G \leftarrow \mathbf{0}}\left(\mathbf{F}^{(l)}\right) = \mathbf{0}$, and $\mathbf{F}_P^{(l)} = g_{\omega_P \leftarrow \mathcal{Q}}\left(\mathbf{F}^{(l)}\right) = \mathbf{F}^{(l)}$, and the overall transformation due to the adapter becomes $\mathbf{F}_A^{(l)} = \mathbf{F}_G^{(l)} \oplus \mathbf{F}_P^{(l)} = \mathbf{0} \ \oplus \ \mathbf{F}^{(l)} = \mathbf{F}^{(l)}$, which shows that our proposed initialization of adapters makes the initial adapter parameter configuration equivalent to an identity function.

Fig. 3 (Right) elaborates on the activation space of the PWC kernel applied on the input feature map, leading to no cross-channel interactions and the same output feature map as input. The proposed initialization scheme makes the adapters behave as an identity function in the beginning, ensuring that source domain knowledge is intact and, thus, no warm-up using the source data is required, unlike approaches such as Niu et al. (2022); Song et al. (2023).

## 3.3 Adapter Parameter Updates and Uncertainty Estimation

For SAPIENT, only the adapter parameters ($\omega$) are trainable and the rest of the network parameters ($\theta_s$) remain frozen as the source domain pre-trained weights. This disentangles the source and target domain knowledge, preventing source domain forgetting, and the learnable $\omega$ parameters can continually acquire knowledge from the dynamically changing target domains. Concretely, source-domain forgetting is prevented by two complementary mechanisms. First, $\theta_s$ is frozen during test-time adaptation, so source-domain representations are preserved regardless of how many target domains are encountered. Second, our identity-equivalent initialization (Section 3.2) ensures that the adapters initially reproduce the frozen network exactly, i.e., $\mathbf{F}_A^{(l)} = \mathbf{F}^{(l)}$, so inserting them causes no interference with source-domain performance. As adaptation proceeds, the adapters learn only residual corrections on top of the frozen features, enabling target-domain adaptation without modifying the source backbone. This separation between frozen source knowledge ($\theta_s$) and learnable target corrections ($\omega$) prevents forgetting while supporting continual adaptation.

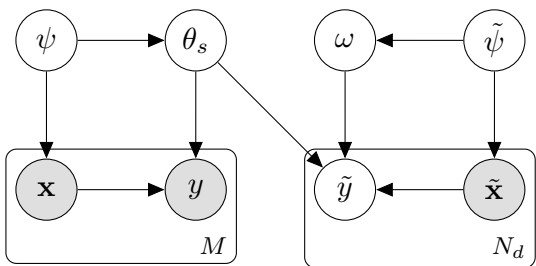

Figure 4: `SAPIENT-B`: Probabilistic model for test-time adaptation with lightweight adapters. $M$ labeled samples are observed during the training time. $N_d$ unlabeled test input samples from a domain $d$ are observed during the test time. Note: $\theta_s$ is the source domain pre-trained model that remains frozen, and only the adapter parameter $\omega$ is adapted during the test-time adaptation. $\psi$ and $\tilde{\psi}$ are the generative parameters for the source and target domain input data, respectively.

Since the learner is agnostic to the change in the domain, we adapt the adapter parameters using examples from a test input batch $\mathbf{x}_b$ from time step $t$ to $t+1$ as $\omega_t \to \omega_{t+1}$, which is done as $\omega_{t+1} = \omega_t - \eta \nabla_\omega \mathcal{L}_\mathcal{U}(f_\omega(\mathbf{x}_b))$, where $\eta$ is the learning rate, $\mathbf{x}_b$ is a test input batch from domain $d$, $f_\omega$ is the model where $\theta_s$ parameters are frozen and only the adapter parameters $\omega$ are learnable. Here, $\mathcal{L}_\mathcal{U}(f_\omega(\mathbf{x}_b))$ is the learning objective defined with respect to adapter parameters $(\omega)$, which can be any unsupervised TTA objective, such as those used in methods such as TENT, EATA, CoTTA, etc. This makes `SAPIENT` highly modular and complementary to these approaches which can benefit from the advantages offered by `SAPIENT`.

Beyond point estimation of adapter parameters, we further propose a Bayesian extension, `SAPIENT-B`, which learns a posterior distribution over the adapter parameters *online* to quantify uncertainty in parameter estimates. The probabilistic graphical model for `SAPIENT-B` is shown in Fig. 4. At inference time, predictions are obtained via Monte-Carlo sampling from this posterior, and averaging across samples yields an ensembling effect that improves robustness (see Table 5).

The parameter-efficiency of `SAPIENT` also facilitates efficiently computing the posterior distribution of network parameters where we follow the idea of keeping the base model frozen with the point estimates of its parameters and computing the posterior only for the adapter parameters (Onal et al., 2024). We would like to emphasize here that, while the idea of adapters has been used for efficient supervised finetuning and continual learning, we leverage adapters for the continual TTA setting where we are required to perform *unsupervised* finetuning of the model at test time. In `SAPIENT-B`, we obtain the adapter parameter distribution, by utilizing SWAG-diagonal-like (Maddox et al., 2019) approximation for the adapter parameter distribution in an *online* continual TTA setting. The running average adapter parameters and the running average of squared adapter parameters are computed from the trajectory of the adapter parameters, as it adapts to incoming batches in an online fashion. These are utilized to obtain the mean and covariance of the Swag-diagonal Gaussian distribution over the adapter parameters. After encountering $T$ batches, let $\omega_1, \omega_2, \ldots, \omega_T$ denote the learning trajectory of the adapter parameters. Then, we obtain the mean of the approximate Gaussian distribution $\omega_\mu$, $\omega_\mu = \frac{1}{T} \sum_{t=1}^{T} \omega_t$. We compute the running average of squared adapter parameters $\overline{\omega^2}$, use it the compute covariance $\omega_\Sigma$ using the mentioned standard identity: $\overline{\omega^2} = \frac{1}{T} \sum_{t=1}^{T} \omega_t^2; \omega_\Sigma = \mathrm{diag}(\overline{\omega^2} - \omega_\mu^2)$. Using these mean and covariances, we obtain the resulting approximate posterior distribution $\mathcal{N}(\omega_\mu, \omega_\Sigma)$. We sample the adapter parameters $\omega$ from this distribution $\mathcal{N}(\omega_\mu, \omega_\Sigma)$, use it along with the remaining frozen network parameters $\theta_s$ to make a prediction $\hat{y}$ for the given test input batch $\mathbf{x}_d$: $\omega \sim \mathcal{N}(\omega_\mu, \omega_\Sigma)$ and $\hat{y} = f_{\theta_s \cup \omega}(\mathbf{x}_d)$. Here, $\theta_s \cup \omega$ denotes the complete parameters of the base network inserted with the adapters between the layers. In order to achieve an *ensembling effect*, we can sample the adapter parameters multiple times, depending on the computational budget, and obtain an averaged prediction.

## 4 Related Work

**Test-Time Adaptation (TTA):** There has been significant recent progress on the problem of test-time adaptation (Liang et al., 2023). Test entropy minimization (TENT) (Wang et al., 2021) adapts the batch-

Table 1: Cityscapes-to-ACDC semantic segmentation online continual test-time adaptation results, measured in terms of mIoU in %. We continually evaluate the four test weather conditions ten times to measure the long-term adaption performance. To conserve space, we only report the continual adaptation results in the first, fourth, seventh, and final rounds. The complete results are provided in Appendix E. The evaluation of all results is conducted utilizing the Segformer-B5 architecture. Note that SAPIENT (shown in the table) only adapts **6.75% adapter parameters**. In contrast, CoTTA requires adapting all (100%) of the base model parameters. [†]ViDA requires source domain data for adapter warm-up initialization and is therefore not strictly source-free. [‡]Continual-MAE results are only shown for the first 3 rounds as reported in their paper.

| Time | $t$ | | | | | | | | | | | | | | | | |
|---|---|---|---|---|---|---|---|---|---|---|---|---|---|---|---|---|---|
| Round | 1 | | | | 4 | | | | 7 | | | | 10 | | | | All |
| Condition | Fog | Night | Rain | Snow | Fog | Night | Rain | Snow | Fog | Night | Rain | Snow | Fog | Night | Rain | Snow | Mean |
| Source | 69.1 | 40.3 | 59.7 | 57.8 | 69.1 | 40.3 | 59.7 | 57.8 | 69.1 | 40.3 | 59.7 | 57.8 | 69.1 | 40.3 | 59.7 | 57.8 | 56.7 |
| BN Adapt | 62.3 | 38.0 | 54.6 | 53.0 | 62.3 | 38.0 | 54.6 | 53.0 | 62.3 | 38.0 | 54.6 | 53.0 | 62.3 | 38.0 | 54.6 | 53.0 | 52.0 |
| TENT-continual | 69.0 | 40.2 | 60.1 | 57.3 | 66.5 | 36.3 | 58.7 | 54.0 | 64.2 | 32.8 | 55.3 | 50.9 | 61.8 | 29.8 | 51.9 | 47.8 | 52.3 |
| ViDA[†] | 71.6 | 43.2 | 66.0 | 63.4 | 70.9 | 44.0 | 66.0 | 63.2 | 72.3 | 44.8 | 66.4 | 62.9 | 72.2 | 45.2 | 65.6 | 62.9 | 61.6 |
| Continual-MAE[‡] | 71.9 | 44.6 | 67.4 | 63.2 | | | — | | | | — | | | | — | | 61.8 |
| CoTTA (100%) | 70.9 | 41.2 | 62.4 | 59.7 | 70.9 | 41.0 | 62.7 | 59.7 | 70.9 | 41.0 | 62.8 | 59.7 | 70.8 | 41.0 | 62.8 | 59.7 | 58.6 |
| % inc. over Round 1 | 0.0 | 0.0 | 0.0 | 0.0 | 0.0 | -0.2 | -0.3 | 0.0 | 0.0 | -0.2 | +0.4 | 0.0 | -0.1 | -0.2 | +0.4 | 0.0 | - |
| SAPIENT (6.75%) | 71.5 | 43.1 | 64.9 | 62.9 | 72.5 | 43.9 | 66.9 | 63.9 | 72.0 | 43.6 | 66.5 | 63.5 | 72.0 | 43.7 | 66.7 | 63.2 | 61.5 |
| % inc. over Round 1 | 0.0 | 0.0 | 0.0 | 0.0 | +1.5 | +0.8 | +2.0 | +1.0 | +0.5 | +0.5 | +1.6 | +0.6 | +0.5 | +0.6 | +1.8 | +0.3 | - |

normalization (BN) parameters utilizing entropy minimization for test data predictions. Schneider et al. (2020) proposes a method to perform test-time adaptation by altering the source domain's batch normalization (BN) statistics using the statistics obtained from the test inputs. EATA (Niu et al., 2022) addresses TTA by employing a weight regularizer; however, it primarily emphasizes on preventing model forgetting of the source knowledge in TTA and does not specifically cater to the challenges associated with forgetting in *continual* TTA. Niu et al. (2023) propose sharpness-aware entropy minimization and batch-agnostic (group or layer) norm for TTA under wild test settings. Chen et al. (2023) utilizes a learnable consistency loss, introducing adaptive parameters after each block, and only updates them during test-time. However, the effectiveness of their proposed adaptive parameters is limited to addressing multi-source and single-source domain generalization tasks for a non-continual setting, and their focus is not on parameter efficiency. Kim et al. (2025) is another very recent work that introduces Buffer modules to perform efficient test-time adaptation.

**Continual Test-time Adaptation:** CoTTA (Wang et al., 2022) addresses the challenge of online continual Test-Time Adaptation (TTA) by utilizing weight averaging and augmentation averaging techniques, as well as randomly restoring parameter values to the source domain model parameters. NOTE (Gong et al., 2022) tackles the challenge of adapting to dynamic target domains by including a normalization layer to handle instances that fall out of distribution and store the simulated i.i.d. data in memory obtained using balanced reservoir sampling. Gan et al. (2023) utilizes image-level visual prompts for adapting to target domains, keeping the source model parameters intact. MECTA (Hong et al., 2023) performs pruning on cache data for back-propagation leading to a reduction in memory requirement. Thus, MECTA is orthogonal to the parameter-efficient approach of making continual TTA efficient with respect to the number of trainable parameters. In a different context than ours, AdaptFormer (Chen et al., 2022) also employ adapters for efficient supervised fine-tuning of pre-trained models. EcoTTA (Song et al., 2023) and BeCoTTA (Lee et al., 2024) utilize meta networks and low-rank experts to adapt the frozen original network to the target domain. However, the main drawbacks of EcoTTA and BeCoTTA are the requirement of source domain training data that is needed in the warm-up process of the meta-networks and low-rank experts, respectively, as well as their inability to quantify uncertainty in the parameters and predictions. Furthermore, we provide a more detailed discussion on comparing SAPIENT to various recent approaches in Appendix A.

## 5 Experiments

To demonstrate the effectiveness of SAPIENT, we perform an extensive evaluation on the Cityscapes-to-ACDC (Cordts et al., 2016; Sakaridis et al., 2021) semantic segmentation benchmark dataset and several classification

Table 2: Error rate (%) results averaged over all corruption types and over 10 diverse corruption orders (highest corruption severity level 5). SAPIENT adapts only a small fraction of the total number of parameters (mentioned inside brackets). CoTTA (100%) means that CoTTA requires adapting all the parameters. [†]ViDA requires source domain data for adapter warm-up initialization and is therefore not strictly source-free.

| Dataset | Metric | Source | BN Adapt | TENT | ViDA[†] | RMT | CoTTA (100%) | SAPIENT (10.92%) |
|---|---|---|---|---|---|---|---|---|
| **ImageNet** | Error (%) | 82.35 | 72.07 | 66.52 | 61.2 | 59.8 | 63.18 | **62.64** |
| **to** | NLL | 5.070 | 3.9956 | 3.6076 | — | — | 3.3425 | **3.3154** |
| **ImageNetC** | Brier | 0.9459 | 0.8345 | 0.8205 | — | — | 0.7681 | **0.7077** |

benchmark datasets (Croce et al., 2021), including ImageNetC, ImageNet3DCC, CIFAR10C, and CIFAR100C. For fairness of comparison, our baselines consist of methods that use the same training objective/mechanism and do not assume access to source domain training data at test time. There are multiple corruptions in a benchmark dataset, and the learner comes across a test input batch remaining agnostic to the information about which domain this batch has come from. For instance, in the semantic segmentation task, the corruptions are the various realistic adverse weather conditions from the Adverse Conditions Dataset (ACDC) (Sakaridis et al., 2021). In the classification task, ImageNetC, CIFAR10C, and CIFAR100C consist of images from 15 different types of image corruptions that can occur due to adverse weather conditions, low light, camera aberration, etc. More details of benchmark datasets are provided in Appendix C.

**Evaluation Metrics:** For evaluation metrics, we follow existing approaches and report the error rate. We also compute negative log-likelihood (NLL) and Brier score to compare the uncertainty estimates of the approaches. Details of all the evaluation metrics are present in Appendix D. For computational complexity and parameter efficiency measures, we use the number of trainable/adaptable parameters along with GPU memory budget and wall-clock time.

### 5.1 Compared Approaches

In order to evaluate the efficacy of SAPIENT, we conduct a comparative analysis of its performance against several (continual) test-time adaptation approaches. *Source* indicates the source domain pre-trained model without any adaptation.

*Pseudo-label* (Lee et al., 2013) utilizes hard pseudo-labels and updates batch normalization parameters using backpropagation. *BN Adapt* (Li et al., 2017; Schneider et al., 2020) only computes batch normalization statistics while keeping all network parameters frozen, including the Batch Norm parameters. *TENT-online* (Wang et al., 2021) denotes the performance of TENT in the setting when the test data arrives continually, but the information about the domain change is accessible. This knowledge about change in the domain makes the learning problem much simpler. Nonetheless, such information regarding a change in the domain may not be readily available in practical situations. *TENT-continual* indicates the performance of TENT in the continual TTA setting, where the domain change information is unavailable. *CoTTA* (Wang et al., 2022) utilizes weight-averaged, augmentation averaged pseudo labels and random restoration of a small part of parameters to the source pre-trained parameters. EATA (Niu et al., 2022) is based on sample-efficient entropy minimization. In Table 6, in addition to these baselines, we also report some additional comparisons of SAPIENT with other recent methods, such as NOTE (Gong et al., 2022) and EcoTTA (Song et al., 2023).

### 5.2 Results

For continual test-time adaptation, Table 1-3 summarizes our results on multiple benchmark datasets where we compare SAPIENT with other methods. For all the experiments with SAPIENT, we use the learning objective and TTA scheme proposed by CoTTA (Wang et al., 2022). In every TTA setting, the model pre-trained on the source dataset is termed the base model ($\theta_s$). CoTTA unfreezes all the model parameters and adapts these parameters during test time. In contrast, SAPIENT adds the proposed lightweight adapters to the pre-trained base model and only adapts the newly added adapter parameters ($\omega$) along with the BN parameters of the base model. Note that the primary objective of SAPIENT is to reduce the parameter update cost while

Table 3: Adaptation results on the online continual test-time adaptation task. The numbers denote the classification error (%) obtained with the highest corruption of severity level 5. TENT-online uses domain information denoted using +. Note that SAPIENT (shown in the table) **only uses (i) 13.61% adapter parameters** for CIFAR10-to-CIFAR10C, and (ii) **6.8% adapter parameters** for CIFAR100-to-CIFAR100C, added to the base model, and only these additional parameters (with BN parameters) are adapted during the test time, keeping the rest of the parameters frozen. In contrast, CoTTA requires adapting all (100%) of the parameters. RMT achieves lower error via full-model symmetric cross-entropy and contrastive learning updates; SAPIENT uses ~7% of trainable parameters while matching CoTTA performance.

| Dataset | Method | Gaussian | shot | impulse | defocus | glass | motion | zoom | snow | frost | fog | bright. | contrast | elastic | pixelate | jpeg | Mean |
|---|---|---|---|---|---|---|---|---|---|---|---|---|---|---|---|---|---|
| CIFAR10C | Source | 72.33 | 65.71 | 72.92 | 46.94 | 54.32 | 34.75 | 42.02 | 25.07 | 41.30 | 26.01 | 9.30 | 46.69 | 26.59 | 58.45 | 30.30 | 43.51 |
| | BN Adapt | 28.08 | 26.12 | 36.27 | 12.82 | 35.28 | 14.17 | 12.13 | 17.28 | 17.39 | 15.26 | 8.39 | 12.63 | 23.76 | 19.66 | 27.30 | 20.44 |
| | Pseudo-label | 26.70 | 22.10 | 32.00 | 13.80 | 32.20 | 15.30 | 12.70 | 17.30 | 17.30 | 16.50 | 10.10 | 13.40 | 22.40 | 18.90 | 25.90 | 19.80 |
| | TENT-online$^+$ | 24.80 | 23.52 | 33.04 | 11.93 | 31.83 | 13.71 | 10.77 | 15.90 | 16.19 | 13.67 | 7.86 | 12.05 | 21.98 | 17.29 | 24.18 | 18.58 |
| | TENT-continual | 24.80 | 20.60 | 28.60 | 14.40 | 31.10 | 16.50 | 14.10 | 19.10 | 18.60 | 18.60 | 12.20 | 20.30 | 25.70 | 20.80 | 24.90 | 20.70 |
| | RMT | 21.90 | 18.60 | 24.10 | 10.80 | 23.60 | 12.00 | 10.40 | 13.00 | 12.40 | 11.40 | 8.30 | 10.10 | 15.20 | 11.30 | 14.60 | 14.50 |
| | CoTTA (100%) | 23.92 | 21.40 | 25.95 | 11.82 | 27.28 | 12.56 | 10.48 | 15.31 | 14.24 | 13.16 | 7.69 | 11.00 | 18.58 | 13.83 | 17.17 | 16.29 |
| | SAPIENT (13.61%) | 24.76 | 21.98 | 26.82 | 11.92 | 28.33 | 12.55 | 10.62 | 15.28 | 14.41 | 13.26 | 7.77 | 12.03 | 19.39 | 14.49 | 18.17 | 16.79 |
| CIFAR100C | Source | 73.00 | 68.01 | 39.37 | 29.32 | 54.11 | 30.81 | 28.76 | 39.49 | 45.81 | 50.30 | 29.53 | 55.10 | 37.23 | 74.69 | 41.25 | 46.45 |
| | BN Adapt | 42.14 | 40.66 | 42.73 | 27.64 | 41.82 | 29.72 | 27.87 | 34.88 | 35.03 | 41.50 | 26.52 | 30.31 | 35.66 | 32.94 | 41.16 | 35.37 |
| | Pseudo-label | 38.10 | 36.10 | 40.70 | 33.20 | 45.90 | 38.30 | 36.40 | 44.00 | 45.60 | 52.80 | 45.20 | 53.50 | 60.10 | 58.10 | 64.50 | 46.20 |
| | TENT-continual | 37.20 | 35.80 | 41.70 | 37.90 | 51.20 | 48.30 | 48.50 | 58.40 | 63.70 | 71.10 | 70.40 | 82.30 | 88.00 | 88.50 | 90.40 | 60.90 |
| | RMT | 38.50 | 34.40 | 35.40 | 26.40 | 32.70 | 27.00 | 25.00 | 27.50 | 27.60 | 30.00 | 24.00 | 25.80 | 27.00 | 25.20 | 28.40 | 29.00 |
| | CoTTA (100%) | 40.09 | 37.67 | 39.77 | 26.91 | 37.82 | 28.04 | 26.26 | 32.93 | 31.72 | 40.48 | 24.72 | 26.98 | 32.33 | 28.08 | 33.46 | 32.48 |
| | SAPIENT (6.8%) | 40.10 | 36.66 | 38.81 | 26.68 | 38.10 | 28.56 | 25.95 | 33.81 | 32.42 | 42.12 | 24.98 | 27.32 | 34.31 | 28.60 | 35.40 | 32.92 |

Table 4: Comparison of SAPIENT applied over TENT, EATA and CoTTA, highlighting the orthogonality of the proposed generic framework. Results for the CIFAR100-to-CIFAR100C benchmark depict the parameter efficiency obtained (**93.2% fewer trainable parameters** compared to CoTTA on the base model).

| Method | Gaussian | shot | impulse | defocus | glass | motion | zoom | snow | frost | fog | bright. | contrast | elastic | pixelate | jpeg | Mean |
|---|---|---|---|---|---|---|---|---|---|---|---|---|---|---|---|---|
| TENT-continual | **37.20** | **35.80** | 41.70 | 37.90 | 51.20 | 48.30 | 48.50 | 58.40 | 63.70 | 71.10 | 70.40 | 82.30 | 88.00 | 88.50 | 90.40 | 60.90 |
| + SAPIENT (6.8% Params) | 41.67 | 39.40 | **41.35** | **26.96** | **40.11** | **28.87** | **26.91** | **33.87** | **33.80** | **39.94** | **26.27** | **29.55** | **34.50** | **32.02** | **40.10** | **34.35** |
| EATA-continual | 41.83 | 40.27 | 42.56 | 27.56 | 41.54 | 29.54 | 27.70 | 34.69 | 34.71 | 41.24 | 26.42 | 30.20 | 35.58 | 32.73 | 40.95 | 35.17 |
| + SAPIENT (6.8% Params) | **41.21** | **38.96** | **41.24** | **26.97** | **41.07** | **29.26** | **27.13** | **33.84** | **34.30** | **39.94** | **26.04** | **30.14** | **34.61** | **31.67** | **39.74** | **34.41** |
| CoTTA (100% Params) | **40.09** | 37.67 | 39.77 | 26.91 | **37.82** | 28.04 | 26.26 | 32.93 | 31.72 | 40.48 | **24.72** | **26.98** | 32.33 | 28.08 | **33.46** | 32.48 |
| + SAPIENT (6.8% Params) | 40.10 | **36.66** | **38.81** | **26.68** | 38.10 | 28.56 | **25.95** | 33.81 | 32.42 | 42.12 | 24.98 | 27.32 | 34.31 | 28.60 | 35.40 | 32.92 |
| + SAPIENT (50.32% Params) | 40.4 | 37.30 | 39.02 | 26.77 | **37.82** | **27.89** | 26.14 | **32.33** | **31.16** | 40.21 | 24.95 | 27.02 | **32.03** | **28.01** | 33.53 | **32.31** |

maintaining the adaptation performance. Since newly added parameters are inserted in between layers, the added adapter modules ensure equal input and output dimensions at the insertion locations of the base model's architecture. Therefore, the fraction/percentage of added parameters may vary depending on the architecture choice of the base model. Refer to Appendix H for architecture-specific adapter locations.

**Cityscapes-to-ACDC:** Following Wang et al. (2022), we consider a Segformer (Xie et al., 2021) pre-trained on the Cityscapes dataset (Cordts et al., 2016) as the base model, and add adapters in between the layers. Further, the model is adapted in test-time over the incoming sequences from ACDC (Sakaridis et al., 2021). Note that the adapters that we propose involve groupwise and pointwise convolution operations. For a transformer-based architecture, we reshape the input stream (flattened) and use a transformed view to pass it through the SAPIENT adapters and perform our proposed initialization (see Section 3.2). Table 1 compares performance with the existing TTA methods. Despite modifying only 6.75% of the parameters in SAPIENT, we surpass the CoTTA baseline by a substantial margin. Moreover, we also compare with ViDA

Table 5: CIFAR100-to-CIFAR100C online continual test-time adaptation task results for the highest corruption of severity level 5. Both `SAPIENT` and `SAPIENT-B` here only use **6.8% adapter parameters** compared to the base model. Note that `SAPIENT` and `SAPIENT-B` here use the TENT learning objective.

| Time | | $t \longrightarrow$ | | | | | | | | | | | | | | | |
|---|---|---|---|---|---|---|---|---|---|---|---|---|---|---|---|---|---|
| **Method** | **Metric** | Gaussian | shot | impulse | defocus | glass | motion | zoom | snow | frost | fog | brightness | contrast | elastic | pixelate | jpeg | Mean |
| TENT | Error % | 37.16 | 35.61 | 41.82 | 37.54 | 51.19 | 48.48 | 49.15 | 58.83 | 62.85 | 71.65 | 70.76 | 82.91 | 88.00 | 91.14 | 94.63 | 61.45 |
| | Brier | 0.51 | 0.52 | 0.63 | 0.60 | 0.82 | 0.82 | 0.8585 | 1.03 | 1.13 | 1.31 | 1.32 | 1.60 | 1.68 | 1.77 | 1.85 | 1.10 |
| | NLL | 1.49 | 1.58 | 2.14 | 2.12 | 3.28 | 3.66 | 4.17 | 5.46 | 6.71 | 8.53 | 9.04 | 14.43 | 14.17 | 16.21 | 17.66 | 7.37 |
| SAPIENT | Error % | 41.67 | 39.4 | 41.35 | 26.96 | 40.11 | 28.87 | 26.91 | 33.87 | 33.8 | 39.94 | 26.27 | 29.55 | 34.5 | 32.02 | 40.1 | 34.35 |
| | Brier | 0.55 | 0.53 | 0.55 | 0.37 | 0.54 | 0.4 | 0.37 | 0.47 | 0.47 | 0.55 | 0.38 | 0.44 | 0.49 | 0.47 | 0.57 | 0.48 |
| | NLL | 1.67 | 1.57 | 1.65 | 1.05 | 1.59 | 1.12 | 1.04 | 1.37 | 1.41 | 1.69 | 1.08 | 1.46 | 1.52 | 1.46 | 1.81 | 1.43 |
| SAPIENT-B | Error % | **40.44** | **35.74** | **37.88** | **25.65** | **37.31** | **28.20** | **25.76** | **32.65** | **31.16** | **37.95** | **25.53** | **29.45** | **32.58** | **29.76** | **37.46** | **32.5** |
| | Brier | **0.54** | **0.48** | **0.5** | **0.36** | **0.51** | **0.39** | **0.36** | **0.46** | **0.44** | **0.53** | **0.36** | **0.42** | **0.46** | **0.43** | **0.53** | **0.45** |
| | NLL | **1.61** | **1.39** | **1.49** | **0.99** | **1.48** | **1.09** | **0.99** | **1.31** | **1.25** | **1.55** | **0.99** | **1.2** | **1.31** | **1.21** | **1.56** | **1.29** |

(Liu et al., 2024b) (61.6 mIoU) and Continual-MAE (Liu et al., 2024a) (61.8 mIoU), which slightly outperform `SAPIENT` (61.5 mIoU) in mean segmentation performance. Critically, ViDA requires source domain data for warmup initialization of its adapters, and Continual-MAE's full-model MAE-based reconstruction carries significantly higher computational overhead. `SAPIENT` achieves comparable segmentation performance in a strictly source-free, parameter-efficient setting using only 6.75% of trainable parameters. Note that in this particular setting (when compared to other classification settings, explained later), there is a recurrence of the target domains that occur cyclically, i.e., Fog → Night → Rain → Snow → Fog → $\cdots$ → Snow. We believe that repeated occurrences of similar domain shifts helps the adaptation performance on longer horizons (40 in number), i.e., starting from round 1, we observe an increase in performance in round 10 (see Table 1, % inc. over round 1). It also points towards a controlled drift of parameters while adaptation, i.e., `SAPIENT` adapting only the adapter parameters helps control the adaptation due to limited freedom in adaptable parameter space, retaining the knowledge learned during initial adaptation on similar distribution shifts, which essentially helps in adaptation when the same distribution occurs again.

**ImageNet-to-ImageNetC:** For this evaluation, we use a pre-trained ResNet-50 as the base model. In this setting, prior works (Wang et al., 2022) report the continual TTA performance over 10 random sequences of the 15 corruptions. To provide a fair comparison, we experiment with `SAPIENT`, considering the same continual setting where the performances are validated for 10 random sequences of corruptions. Table 2 shows the performance over 10 different runs. We observe that with only 10.92% of added trainable adapter parameters, `SAPIENT` achieves an improvement in terms of error rate over CoTTA (Wang et al., 2022) (from 63.18% to 62.64%). This highlights that the parameter update cost can be significantly reduced for existing approaches with no performance drop. We additionally include ViDA (Liu et al., 2024b) (61.2%) and RMT (Döbler et al., 2023) (59.8%), which outperform `SAPIENT` (62.64%) on ImageNetC; however, ViDA requires source data for adapter warmup and RMT updates all model parameters, whereas `SAPIENT` achieves competitive accuracy using only ∼11% trainable parameters with no source data access.

**CIFAR10-to-CIFAR10C:** We use pre-trained WideResNet-28 (Zagoruyko and Komodakis, 2016) as a base model for experiments on CIFAR10. For `SAPIENT`, we add lightweight adapters with only 13.6% of the number of the base model's parameters. Table 3 reports the continual TTA error rates (Appendix G contains results on Brier score and NLL) of all the methods on CIFAR10C, where various corruptions occur continually in a sequence of mini-batches with a batch size of 200. With 86% reduction in the number of trainable/adaptable parameters, `SAPIENT` achieves a similar average performance with a drop of 0.5% in terms of the mean error rate compared to CoTTA.

**CIFAR100-to-CIFAR100C:** For CIFAR100C, we use pre-trained ResNeXt-29 (Xie et al., 2017) as the base model. Table 3 report the error rates (Appendix G contains results on Brier score and NLL) over the sequence of corruptions. `SAPIENT` adds only 6.8% adapter parameters to the pre-trained ResNeXt-29 for adaptation

Table 6: Comparison of `SAPIENT` with other existing TTA methods in terms of the error rate (%) on CIFAR10-to-CIFAR10C and ImageNet-to-ImageNetC datasets. Note that `SAPIENT` uses the learning objective and TTA scheme proposed by CoTTA, and all the comparisons are made in a continual setting.

| Method | Source | BN-Adapt | TENT | EATA | NOTE | ECoTTA | CoTTA | SAPIENT |
|---|---|---|---|---|---|---|---|---|
| **Source Free** | ✓ | ✓ | ✓ | ✗ | ✓ | ✗ | ✓ | ✓ |
| **Adapter** | ✗ | ✗ | ✗ | ✗ | ✗ | ✓ | ✗ | ✓ |
| **Param. Uncer.** | ✗ | ✗ | ✗ | ✗ | ✗ | ✗ | ✗ | ✓ |
| **CIFAR10C** | 43.51 | 20.44 | 20.7 | 18.6 | 20.2 | 16.8 | **16.29** | 16.79 |
| **ImageNetC** | 82.35 | 72.07 | 66.52 | 63.8 | - | 63.4 | 63.18 | **62.64** |

during inference. The results show that `SAPIENT` achieves a mean error rate of 32.92% with a reduction of $\sim$ 93% in terms of the number of trainable parameters. Adapting the entire model parameters, CoTTA achieves an improvement of only 0.34% over `SAPIENT` in terms of average error rate. Moreover, as observed for some corruptions, like shot, impulse, defocus, and zoom, `SAPIENT` achieves a marginal improvement over CoTTA with significant savings in the parameter update cost. We additionally compare with RMT (Döbler et al., 2023), which achieves lower mean error rates (14.5% on CIFAR10C and 29.0% on CIFAR100C) through full-model symmetric cross-entropy and contrastive learning updates. `SAPIENT`, using $\sim$7% of trainable parameters, achieves comparable performance to CoTTA while offering significant parameter efficiency.

Overall, our detailed results across various benchmarks highlight that `SAPIENT` achieves a comparable performance with huge efficiency in the number of trainable parameters. Fig. 2 highlights the comparable performance achieved with significant efficiency using `SAPIENT` over CoTTA. In Table 6, we report a broader comparison of SAPIENT with other continual TTA methods. Note that SAPIENT does not claim to outperform all existing methods; rather, the goal of this comparison is to show that SAPIENT achieves *competitive* performance relative to the evaluated baselines while simultaneously being source data free and requiring substantially fewer trainable parameters using adapters. For more details about the hyperparameters, refer to Appendix B. For experiments on ImageNet-to-ImageNet3DCC, refer to Appendix F.

### 5.3 Learning Distribution over Adapter Parameters

Our proposed `SAPIENT` adapters have drastically fewer parameters than the base network, making Bayesian learning in the low dimensional adapter parameter space much more efficient. We perform SWAG-diagonal-like (Maddox et al., 2019) approximation for learning the adapter parameter distribution. In Table 5, we use `SAPIENT-B` to denote the Bayesian extension of `SAPIENT`. In the experiments, we perform ensembling with 10 samples of the adapter parameters. We observe superior performance in terms of error, as well as uncertainty based metrics, such as Brier score and NLL, thereby demonstrating the robustness of `SAPIENT-B`. Note that even though here we have used SWAG (Maddox et al., 2019) as the posterior approximation algorithm, it is also possible to use other approximate Bayesian inference methods, such as Laplace approximation (Daxberger et al., 2021) or variational inference.

## 6 Discussion

**Flexibility in Parameter Efficiency:** The overall objective of the test-time adaptation methods is to increase the usage of existing methods in the real-world changing environment over time, making the models more robust towards domain shifts when deployed in the wild. However, it is imperative that a proposed method does not compromise upon the predictive performance. `SAPIENT` provides flexibility in choosing the desired number of additional adapter parameters for a task. To validate if the same performance can be achieved by adding more adapter parameters ($\omega$), we experiment with the `SAPIENT` setting, where we increase the trainable number of parameters by adding more adapters to the base model. We experiment with multiple settings where we add different number of parameters. Table 14 in the Appendix highlights the performance comparison along with the parameter comparison in detail. As observed from the results, increasing the number of adapter parameters does help boost the performance and making the trainable parameters 57.14%

Table 7: Time and memory budget requirements for CIFAR100-to-CIFAR100C upon varying number of trainable parameters, along with the error rate (%).

| | Adapt. Params | Mem. (MB) | TTA Time (secs) \| Error (%) | | |
| --- | --- | --- | --- | --- | --- |
| | | | CoTTA | TENT | EATA |
| BN Params | 0.37% | 2890.56 | 110.80 \| 34.70 | 18.60 \| 60.90 | 28.06 \| 35.17 |
| All Params | 100% | 5693.27 | 149.70 \| 32.48 | 28.53 \| 33.64 | 33.80 \| 33.89 |
| SAPIENT | 6.80% | 3311.87 | 136.14 \| 32.92 | 21.20 \| 32.92 | 28.60 \| 34.41 |

of the base model achieves 32.65% mean error rate, which is very close to CoTTA which requires retraining all (i.e., 100%) the parameters. Moreover, we also observe that adding a similar number of trainable parameters using SAPIENT (101.29% of the base model) helps achieve marginal performance improvement over the CoTTA baseline (32.5% to 32.31% mean error rate).

**Orthogonality with existing TTA approaches:** As SAPIENT emphasizes parameter efficiency, the proposed method and the learning objective $\mathcal{L}_{\mathcal{U}}(f_\omega())$ defined with respect to adapter parameters ($\omega$) is generic and can be any unsupervised test-time adaptation loss, making it orthogonal to existing approaches. To validate the orthogonality with existing LTTA approaches, we perform another set of experiments where we combine SAPIENT with the learning objectives proposed by TENT, EATA, and CoTTA.

Table 4 reports the results for SAPIENT using the learning objective of TENT, EATA, and CoTTA. Note that TENT and EATA propose adapting only BN parameters (0.37%) during TTA, whereas CoTTA adapts the entire network weights (100%).

Table 7 depicts a decrease in inference/TTA time and memory budget requirement, along with the error rate comparison between various architecture settings and different continual TTA methods for the CIFAR100-to-CIFAR100C dataset. Even though updating *BN Params* adapts a minimal number of parameters, SAPIENT with only 6.80% adaptable parameters outperforms *BN Params* significantly and performs comparably with *All Params* version, consistently across CoTTA, TENT, and EATA. Thus, SAPIENT provides an advantage of adapting a minuscule percentage of parameters to achieve similar/better performance based on the available memory/time budget, making it more practical and flexible for real-life deployment of the TTA models. Further, Table 5 shows that introducing SAPIENT-B adapters (a Bayesian extension of SAPIENT) to TENT can improve the uncertainty quantification as well as the error rates. Similar to TENT, SAPIENT-B adapters can be introduced to other TTA approaches. We further validate the orthogonality of SAPIENT with source-anchoring based methods by combining it with SANTA (Chakrabarty et al., 2023); the results are provided in Appendix G and Table 15.

## 7 Conclusion

We propose a generic framework for making adaptation efficient during test time and introduce SAPIENT: continual teSt-time **A**da**P**tation v**I**a lightw**E**ight plug-a**N**d-play adap**T**ers. SAPIENT uses efficient adapters, which can be trained at test-time (using unlabeled test inputs) to improve the performance under domain shifts. SAPIENT offers two key advantages: making the adaptation parameter efficient and keeping the source knowledge intact. With the proposed initialization scheme, SAPIENT also removes the dependency on the source dataset at the adaptation time (required by other recent methods), making the proposed adapters compatible with the full-test-time adaptation setting. SAPIENT requires substantially ($\sim$**90%**) fewer trainable parameters to achieve competitive or comparable performance relative to the evaluated continual TTA methods, resulting in faster adaptation and inference during test time. In addition, the ability to estimate the uncertainty in the adapter parameters using a Bayesian approach is another appealing advantage of SAPIENT which is not present in existing continual TTA methods. Moreover SAPIENT can be seamlessly used as a plug-and-play manner with other continual TTA methods to improve them further. The proposed adapters and initialization scheme will help provide parameter control to the test-time adaptation approaches and make them more efficient for real-world use cases.

## Broader Impact Statement

This paper presents a technique to enable robust deployment of a deep neural network - a source model - in the "in-the-wild" settings where the distribution of the test inputs can potentially change over time and the source model needs to be *unsupervisedly* adapt itself to such changes. The proposed method is used for supervised learning problems (e.g., classification, segmentation, etc) and we do not anticipate any use of the model for the creation of harmful content (since it is not a generative AI model). However, since the model is adapted at test-time using test inputs, there may be a potential risk of an attacker influencing the model using malicious test inputs. That being said, one of the appealing aspects of our model is that the adaptation only happens for the adapter modules while the source model remains preserved, which mitigates the aforementioned risk. In addition, the proposed continual test-time adaptation method is likely to result in improved performance in domains such as computer vision and NLP. It does raise some possibility of both positive and negative uses. This paper presents work whose goal is to advance the field of Machine Learning. There are many potential societal consequences of our work, none of which we feel must be specifically highlighted here.

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

# Appendix

## A  Other Related Works

LoRA (Hu et al., 2021), a popular parameter-efficient fine-tuning (PEFT) approach for supervised adaptation of large language models, decomposes weight updates into low-rank matrices initialized from random projections. Unlike `SAPIENT`, LoRA requires labeled source data for fine-tuning and does not admit an identity-equivalent initialization, making it incompatible with the unsupervised, source-free continual TTA setting. Table 8 summarizes how `SAPIENT` differs from these closely related adapter-based approaches. Several recent works

Table 8: Comparison of `SAPIENT` with closely related adapter-based and PEFT methods across key properties relevant to the continual TTA setting.

| Method | Source Data Free | Uncertainty (UQ) | Plug-and-Play | Continual TTA |
|---|---|---|---|---|
| LoRA (Hu et al., 2021) | ✗ | ✗ | ✗ | ✗ |
| EcoTTA (Song et al., 2023) | ✗ | ✗ | ✗ | ✓ |
| BeCoTTA (Lee et al., 2024) | ✗ | ✗ | ✗ | ✓ |
| **SAPIENT (Ours)** | ✓ | ✓ | ✓ | ✓ |

have further expanded the landscape of continual TTA. RMT (Döbler et al., 2023) employs a robust mean teacher mechanism but updates the full model, making it orthogonal to and potentially complementary with our parameter-efficient adapter approach. ViDA (Liu et al., 2024b) introduces homeostatic visual domain adapters for continual TTA; however, its best-performing variant relies on source domain data to configure the adapters during a warmup phase, a dependency that `SAPIENT`'s identity-equivalent initialization explicitly eliminates. Continual-MAE (Liu et al., 2024a) proposes a masked-autoencoder-based adaptive distribution modeling approach, which differs fundamentally from SAPIENT in both adaptation paradigm and design objective, as it does not target parameter efficiency.

## B  Hyperparameter Setting

We use PyTorch (Paszke et al., 2019) to develop and train our architecture and RobustBench (Croce et al., 2021) for the various pre-trained architectures used in the experiments. We use the existing set of pre-trained models (base models) that are widely used for TTA experiments. We run all the experiments over the NVIDIA A40 GPU. For hyperparameters, we use the available set of hyperparameters proposed by the TTA approaches; for example, for using the TTA mechanism proposed by CoTTA, we use hyperparameters provided by CoTTA. For a fair comparison, we use the same optimizers (Adam (Kingma and Ba, 2014) and

SGD) reported by previous TTA baselines. As we decrease the number of parameters by a significant margin, we tune the learning rate for various settings.

For Cityscapes-to-ACDC experiments, we employ the Adam optimizer (Kingma and Ba, 2014) with a learning rate of $0.0006/8$ and a $\beta$ value of 0.9. No weight decay is applied. Following CoTTA, the teacher model weights are updated using an exponential moving average, utilizing the student model weights with a decay factor of $\alpha = 0.999$ with the reset rate of 0.01%.

For CIFAR10-to-CIFAR10C experiments, we use Adam optimizer (Kingma and Ba, 2014) with a learning rate of 0.00125, and $\beta = 0.9$ with no weight decay. We follow CoTTA for the mean teacher parameter and update the weights of the teacher model by exponential moving average using the student model weights using $\alpha = 0.999$, but without any resetting, i.e., reset rate of 0%.

For CIFAR100-to-CIFAR100C experiments, we use a learning rate of 0.0015 for the Adam optimizer with $\beta = 0.9$ and no weight decay. For the mean teacher parameter, we follow CoTTA and update the weights of the teacher model by exponential moving average using the student model weights using $\alpha = 0.999$ with a reset rate of 1%.

In the ImageNet-to-ImageNetC experiments, we use a stochastic gradient descent (SGD) optimizer with a learning rate of 0.04, momentum of 0.9, and no weight decay. We follow CoTTA for the mean teacher weight parameter and update teacher model weights by exponential moving average using student model weights with $\alpha = 0.999$, and a reset rate of 0.1%.

The ImageNet-to-ImageNet3DCC experiments use an SGD optimizer with a learning rate of 0.03, a momentum of 0.9, with no weight decay. We follow CoTTA for the mean teacher weight parameter. The teacher model weights are updated by exponential moving average, utilizing the student model weights with $\alpha = 0.999$ and a reset rate of 0.1%.

## C   Benchmark Datasets

### C.1   Semantic Segmentation Task

Cityscapes-to-ACDC is a continual semantic segmentation test-time adaptation benchmark task proposed by CoTTA (Wang et al., 2022) that simulates the realistic continual drift in data distribution. The segmentation model trained on the Cityscapes dataset (Cordts et al., 2016) is used as the source domain pre-trained model. The target domains comprise images from several scenarios sourced from the Adverse Conditions Dataset (ACDC) (Sakaridis et al., 2021). The ACDC dataset consists of identical semantic classes as the Cityscapes dataset. The corruptions in the target domain are the four distinct adverse weather conditions in ACDC: Fog, Night, Rain, and Snow. Following CoTTA (Wang et al., 2022), we compare the approaches using the identical default sequence of corruptions. During the test-time adaptation, we utilize 400 unlabeled images from each adverse weather condition. We replicate real-life situations where one may encounter similar conditions multiple times and assess the impact of the compared approaches on memory retention of previous conditions by repeating the same sequence group of four conditions for ten rounds. Thus, we have a total of 40 conditions in ten rounds: Fog → Night → Rain → Snow → Fog → Rain · · · ·. Moreover, this offers an assessment of the long-term effectiveness of the adaptation.

### C.2   Classification Task

Multiple corruptions are introduced to the standard CIFAR10 and CIFAR100 (Krizhevsky, 2009) datasets to get CIFAR10C and CIFAR100C datasets are corrupted versions, respectively. Both ImageNetC (Hendrycks and Dietterich, 2019) and ImageNet3DCC (Kar et al., 2022) are corrupted versions of the standard ImageNet (Deng et al., 2009) dataset.

The CIFAR10C and CIFAR100C datasets both consist of 10,000 images for every corruption type, resulting in a total of 150,000 images for each dataset. The ImageNetC dataset consists of 50,000 images for each corruption class. The CIFAR10C, CIFAR100C, and ImageNet-C datasets comprise a total of 15 distinct types of corruptions, with an additional 4 types designated for validation purposes. Every corruption consists

of five distinct levels of severity. The different kinds of corruption, accompanied by concise explanations, are outlined below:

1. Gaussian noise: frequently observed in situations characterized by low illumination levels

2. Shot noise: electrical noise that arises from the discrete character of light

3. Impulse noise: color counterpart of salt-and-pepper noise and can potentially occur owing to bit errors

4. Defocus blur: occurs when an image is captured with an improper focus

5. Frosted glass blur: occurs when an image is seen through a window having frosted glass

6. Motion blur: a phenomenon that arises when a camera undergoes rapid movement

7. Zoom blur: occurs when a camera rapidly moves toward an object

8. Snow: a form of precipitation that visibly obscures the object of interest

9. Frost: happens when ice crystals stick to windows

10. Fog: the presence of fog in the environment causes objects to be obscured from view; this effect is generated using the diamond-square algorithm

11. Brightness: subject to change in accordance with the intensity of sunlight

12. Contrast: influenced by the lighting conditions and the color of the photographed object

13. Elastic transformations: stretching or contracting of small image portions

14. Pixelation: occurs when a low-resolution image is upsampled

15. JPEG: lossy image compression method that leads to the formation of compression artifacts

The ImageNet 3D Common Corruptions (ImageNet3DCC) dataset, as proposed in a recent work by Kar et al. (2022), utilizes the scene geometry for transformations, leading to the generation of corruptions more closely resembling real-world scenarios. The Imagenet3DCC dataset consists of 50,000 images for every form of corruption included within the dataset. It comprises a total of 12 different kinds of corruption, each characterized by 5 degrees of severity. The instances of corruption can be categorized as follows:

1. Near focus: altering the focus region to the nearby section of the scene in a random manner

2. Far focus: introduce random alterations in the focus to encompass the far portion of the scene

3. Bit error: attributed to the presence of imperfections in the video transmission channel

4. Color quantization: reduces the bit depth of an RGB image

5. Flash: occurs when a light source is placed in close proximity to the camera

6. Fog 3D: produced by utilizing a conventional optical model for fog

7. H.265 ABR: H.265 codec in conjunction with the Average Bit Rate control mode for compression purposes

8. H.265 CRF: H.265 codec for compression purposes, specifically employing the Constant Rate Factor (CRF) control mode

9. ISO noise: refers to the presence of noise in an image, which follows a Poisson-Gaussian distribution

10. Low-light: simulated by lowering pixel intensities and addition of Poisson-Gaussian distributed noise

11. XY-motion blur: refers to the blur when the primary camera is in motion along the XY-plane of the picture

12. Z-motion blur: occurs when the primary camera is moving along the Z-axis of the image

The purpose of developing these datasets is to provide standardized benchmarks for evaluating the robustness of classification models.

## D Evaluation Metrics

For a given dataset, assume $D = \{x_n, y_n\}_{n=1}^{N}$, with $y_n$ to be the true label (in one-hot representation, i.e., $y_{ni} = 1$ if $i$ is the true class label, else $y_{ni} = 0$) of $x_n$, and $y'_n$ to be the prediction by the model.

### D.1 Error

The definition of average error rate is as follows:

$$\text{Error} = \frac{1}{N} \sum_{n=1}^{N} \mathbb{I}(y'_n \neq y_n). \tag{2}$$

Here $\mathbb{I}()$ denotes the indicator function.

### D.2 Brier Score

The average Brier score (Brier, 1950) is given by the following:

$$\text{Brier score} = \frac{1}{N} \sum_{n=1}^{N} \sum_{i=1}^{D} (y'_{ni} - y_{ni})^2. \tag{3}$$

### D.3 Negative Log-Likelihood

We define average negative log-likelihood (NLL) as:

$$\text{NLL} = -\frac{1}{N} \sum_{n=1}^{N} \sum_{i=1}^{D} (y_{ni} \log y'_{ni}). \tag{4}$$

## E Cityscapes-to-ACDC Results

Table 9 reports the complete experimental results for Cityscapes-to-ACDC semantic segmentation online continual test-time adaptation task for every adverse weather condition across the ten rounds.

## F ImageNet-to-ImageNet3DCC Results

Table 10 reports the experimental results for ImageNet-to-ImageNet3DCC image classification online continual test-time adaptation task. For this evaluation, we use the same architecture as that of ImageNetC experiments with a pre-trained ResNet-50 and the same number of added adaptable parameters (10.92%). Table 10 highlights the performance over 10 random orders of corruptions. We observe that with only 10.92% of added adaptable parameters, SAPIENT achieves a comparable average error rate of 60.47% compared to an average error rate of 59.91% for CoTTA (with all parameters), with only 0.56% performance drop.

Table 9: Cityscapes-to-ACDC semantic segmentation online continual test-time adaptation results, measured in terms of mIoU in %. We continually evaluate the four test weather conditions ten times to measure the long-term adaption performance. The evaluation of all results is conducted utilizing the Segformer-B5 architecture. Note that SAPIENT (shown in the table) only uses **6.75% adapter parameters** added to the base model, and only these additional parameters (with BN parameters) are adapted during the test time, keeping the rest of the parameters frozen. In contrast, CoTTA requires adapting all (100%) of the parameters. [†]ViDA requires source domain data for adapter warm-up initialization and is therefore not strictly source-free. [‡]Continual-MAE results are only available for the first 3 rounds of the target domain sequence as reported in their paper.

| Time | t →→→→→→→→→→→→→→→→→→→→→→→→→→→→→→→→→→→→→→→→→→→→→→→→→ | |
|---|---|---|
| **Condition** | Fog Night Rain Snow \| Fog Night Rain Snow \| Fog Night Rain Snow \| Fog Night Rain Snow \| Fog Night Rain Snow | cont. |
| **Round** | 1 \| 2 \| 3 \| 4 \| 5 | cont. |
| **Source** | 69.1 40.3 59.7 57.8 \| 69.1 40.3 59.7 57.8 \| 69.1 40.3 59.7 57.8 \| 69.1 40.3 59.7 57.8 \| 69.1 40.3 59.7 57.8 | cont. |
| **BN Adapt** | 62.3 38.0 54.6 53.0 \| 62.3 38.0 54.6 53.0 \| 62.3 38.0 54.6 53.0 \| 62.3 38.0 54.6 53.0 \| 62.3 38.0 54.6 53.0 | cont. |
| **TENT-continual** | 69.0 40.2 60.1 57.3 \| 68.3 39.0 60.1 56.3 \| 67.5 37.8 59.6 55.0 \| 66.5 36.3 58.7 54.0 \| 65.7 35.1 57.7 53.0 | cont. |
| **CoTTA** | 70.9 41.2 62.4 59.7 \| 70.9 41.1 62.6 59.7 \| 70.9 41.0 62.7 59.7 \| 70.9 41.0 62.7 59.7 \| 70.9 41.0 62.8 59.7 | cont. |
| **ViDA[†]** | 71.6 43.2 66.0 63.4 \| 73.2 44.5 67.0 63.9 \| 73.2 44.6 67.2 64.2 \| 70.9 44.0 66.0 63.2 \| 72.0 43.7 66.3 63.1 | cont. |
| **Continual-MAE[‡]** | 71.9 44.6 67.4 63.2 \| 71.7 44.9 66.5 63.1 \| 72.3 45.4 67.1 63.1 \| — \| — | cont. |
| **SAPIENT** | **71.5 43.1 64.9 62.9** \| **72.8 44.8 66.9 63.7** \| **72.7 45.1 67.0 63.9** \| **72.5 43.9 66.9 63.9** \| **72.2 42.6 66.7 63.9** | cont. |

| **Round** | 6 \| 7 \| 8 \| 9 \| 10 | Mean |
|---|---|---|
| **Source** | 69.1 40.3 59.7 57.8 \| 69.1 40.3 59.7 57.8 \| 69.1 40.3 59.7 57.8 \| 69.1 40.3 59.7 57.8 \| 69.1 40.3 59.7 57.8 | 56.7 |
| **BN Adapt** | 62.3 38.0 54.6 53.0 \| 62.3 38.0 54.6 53.0 \| 62.3 38.0 54.6 53.0 \| 62.3 38.0 54.6 53.0 \| 62.3 38.0 54.6 53.0 | 52.0 |
| **TENT-continual** | 64.9 34.0 56.5 52.0 \| 64.2 32.8 55.3 50.9 \| 63.3 31.6 54.0 49.8 \| 62.5 30.6 52.9 48.8 \| 61.8 29.8 51.9 47.8 | 52.3 |
| **CoTTA** | 70.9 41.0 62.8 59.7 \| 70.9 41.0 62.8 59.7 \| 70.9 41.0 62.8 59.7 \| 70.8 41.0 62.8 59.7 \| 70.8 41.0 62.8 59.7 | 58.6 |
| **ViDA[†]** | 72.2 44.0 66.6 62.9 \| 72.3 44.8 66.4 62.9 \| 72.1 45.1 66.2 62.9 \| 71.9 43.5 66.3 62.9 \| 72.2 45.2 65.6 62.9 | 61.6 |
| **Continual-MAE[‡]** | — \| — \| — \| — \| — | 61.8 |
| **SAPIENT** | **72.2 43.2 66.8 63.8** \| **72.0 43.6 66.5 63.5** \| **72.0 43.3 66.7 63.4** \| **71.6 43.0 66.7 63.6** \| **72.0 43.7 66.7 63.2** | 61.5 |

Table 10: Error rate (%) results averaged over all corruption types and over 10 diverse corruption orders (highest corruption severity level 5). SAPIENT adapts only a small fraction of the total number of parameters (mentioned inside brackets). CoTTA (100%) means that CoTTA requires adapting all the parameters.

| Dataset | Metric | Source | BN Adapt | TENT | CoTTA (100%) | SAPIENT (10.92%) |
|---|---|---|---|---|---|---|
| **ImageNet** | Error (%) | 69.21 | 67.32 | 95.93 | **59.91** | 60.47 |
| **to** | NLL | 3.9664 | 3.7163 | 19.0408 | **3.2636** | 3.3018 |
| **ImageNet3DCC** | Brier | 0.8080 | 0.7872 | 1.8031 | **0.7270** | 0.7365 |

# G   Additional Results

For a fair comparison with existing TTA methods, we report all the metrics results corresponding to Table 3 in Table 11 and Table 12, respectively. Overall, we observe that similar performance can be achieved with a significant reduction in trainable parameters.

**Comparison with other TTA methods:** The main focus of this work is to design parameter-efficient adapters that are well-suited for continual test-time adaptation (TTA). All comparisons conducted in our work are based on the mechanism proposed by CoTTA. In addition, for completeness, we also report a comparative analysis of the performance of our approach with some other recently proposed continual TTA approaches.

Table 6 provides a comparison with other existing methods. Since different methods show results on various architectures, we typically use the standard architectures and report the numbers from the paper corresponding to the same architecture and dataset. Note that another recent work, EcoTTA (Song et al., 2023) proposes to include meta-network modules to the base model for reducing the activation and memory cost in TTA methods. Adding more parameters in EcoTTA still requires a warm-up phase, making it dependent on the availability of the source dataset. In contrast, our approach SAPIENT removes this dependency by proposing

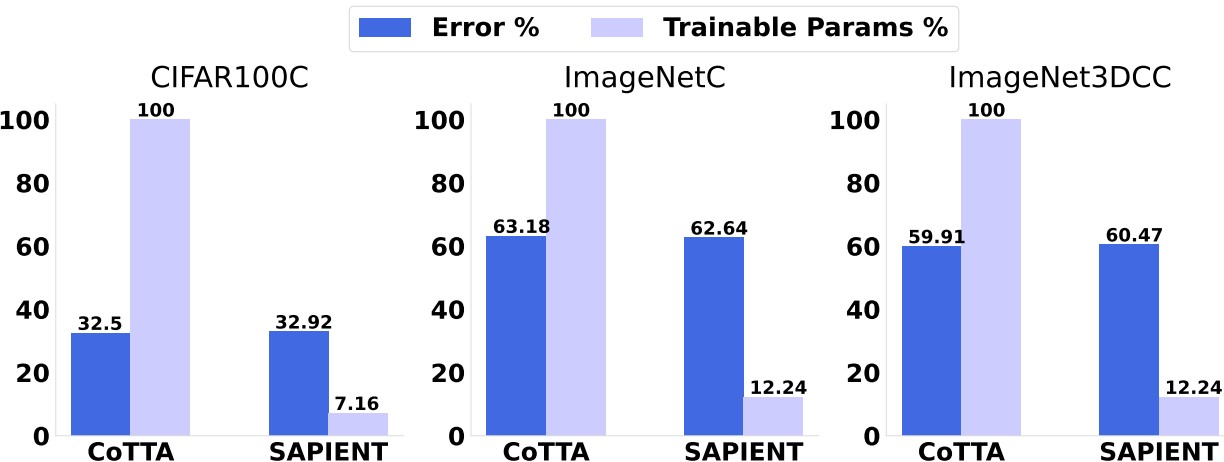

Figure 5: Error rate (%) for CIFAR100-to-CIFAR100C, ImageNet-to-ImageNetC and ImageNet-to-ImageNet3DCC along with % of trainable parameters. Higher is better for accuracy, whereas lower is better for the trainable parameter percentage.

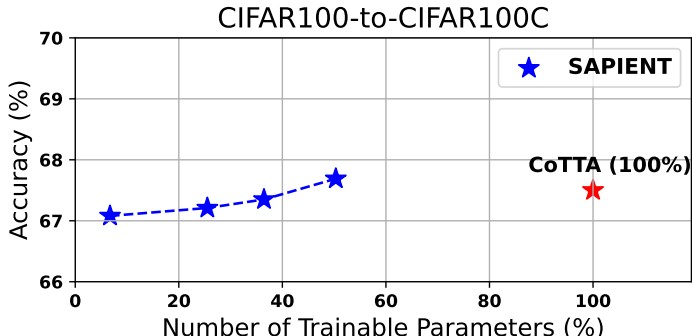

Figure 6: Accuracy rate (%) on CIFAR100C over different percentages of added parameters in `SAPIENT`. The number of trainable parameters comparison is shown, keeping CoTTA as a baseline (i.e., 100%). Refer to Appendix Table 14 for the exact number of parameters with more details.

adapter designs compatible with identity transformation for initialization, making it generic for all TTA methods. Note that EcoTTA establishes the baseline adaptation performances in the segmentation task using a different CNN-based architecture (DeepLabV3Plus-ResNet-50). In our experiments, we consider choosing Segformer, following Wang et al. (2022). Moreover, experiments on Segformer also highlight the applicability of `SAPIENT` across various kinds of architectures (transformers, CNNs).

**Orthogonality with existing TTA approaches:** TENT and EATA only make use of BN parameters in their proposed approach. When compared to the BN parameters adaptable version of both TENT and EATA, TENT+`SAPIENT` and EATA+`SAPIENT` both achieve significant performance improvement; we speculate the primary reason to be the usage of more parameter space for adaptation without losing the source model weights. Hence, to evaluate the dependence throughout the adaptable parameter space, we create an additional setting in which we update the complete model parameters (100% trainable parameters) and report the findings. Similarly, for comparison with CoTTA, we add an additional setting of adapting only BN parameters. Experimental results in Table 13 show that adapting the entire model parameters does help boost the performance by a significant margin; however, it loses the proxy for source knowledge as all the parameters are now updated. We discover that `SAPIENT` can achieve a similar performance improvement with

Table 11: The table shows all the metrics results obtained for the CIFAR10-to-CIFAR10C online continual test-time adaptation task for the highest corruption of severity level 5 corresponding to the results obtained for `SAPIENT` reported in Table 3. `SAPIENT` here only uses 13.61% adapter parameters compared to the base model. Note that `SAPIENT` here uses the learning objective and TTA scheme proposed by CoTTA.

| Method | Metric | Gaussian | shot | impulse | defocus | glass | motion | zoom | snow | frost | fog | brightness | contrast | elastic | pixelate | jpeg | Mean |
|---|---|---|---|---|---|---|---|---|---|---|---|---|---|---|---|---|---|
| Source | Error % | 72.33 | 65.71 | 72.92 | 46.94 | 54.32 | 34.75 | 42.02 | 25.07 | 41.30 | 26.01 | 9.30 | 46.69 | 26.59 | 58.45 | 30.30 | 43.51 |
| | Brier | 1.29 | 1.16 | 1.21 | 0.79 | 0.93 | 0.59 | 0.71 | 0.42 | 0.72 | 0.44 | 0.15 | 0.77 | 0.44 | 1.02 | 0.50 | 0.74 |
| | NLL | 6.46 | 5.61 | 5.47 | 2.74 | 3.84 | 2.09 | 2.51 | 1.51 | 3.15 | 1.53 | 0.48 | 2.69 | 1.38 | 4.67 | 1.65 | 3.05 |
| BN | Error % | 28.08 | 26.12 | 36.27 | 12.82 | 35.28 | 14.17 | 12.13 | 17.28 | 17.39 | 15.26 | 8.39 | 12.63 | 23.76 | 19.66 | 27.30 | 20.44 |
| | Brier | 0.46 | 0.43 | 0.59 | 0.20 | 0.57 | 0.23 | 0.19 | 0.28 | 0.28 | 0.24 | 0.13 | 0.20 | 0.38 | 0.32 | 0.45 | 0.33 |
| | NLL | 1.46 | 1.32 | 1.90 | 0.57 | 1.76 | 0.64 | 0.54 | 0.82 | 0.82 | 0.71 | 0.36 | 0.57 | 1.14 | 0.92 | 1.38 | 0.99 |
| TENT | Error % | 24.80 | 20.48 | 28.49 | 14.84 | 31.78 | 16.97 | 16.66 | 21.97 | 20.97 | 20.92 | 14.76 | 19.91 | 27.56 | 23.89 | 31.01 | 22.33 |
| | Brier | 0.42 | 0.35 | 0.50 | 0.26 | 0.56 | 0.30 | 0.30 | 0.40 | 0.39 | 0.38 | 0.27 | 0.37 | 0.51 | 0.44 | 0.58 | 0.40 |
| | NLL | 1.41 | 1.33 | 2.10 | 1.11 | 2.61 | 1.52 | 1.65 | 2.34 | 2.43 | 2.42 | 1.76 | 2.48 | 3.21 | 2.97 | 4.17 | 2.23 |
| CoTTA | Error % | 23.92 | 21.40 | 25.95 | 11.82 | 27.28 | 12.56 | 10.48 | 15.31 | 14.24 | 13.16 | 7.69 | 11.00 | 18.58 | 13.83 | 17.17 | 16.29 |
| | Brier | 0.36 | 0.33 | 0.38 | 0.18 | 0.40 | 0.19 | 0.16 | 0.23 | 0.21 | 0.20 | 0.11 | 0.16 | 0.27 | 0.20 | 0.25 | 0.24 |
| | NLL | 0.92 | 0.85 | 0.88 | 0.43 | 0.93 | 0.46 | 0.37 | 0.55 | 0.50 | 0.46 | 0.26 | 0.36 | 0.60 | 0.45 | 0.56 | 0.57 |
| SAPIENT | Error % | 24.76 | 21.98 | 26.82 | 11.92 | 28.33 | 12.55 | 10.62 | 15.28 | 14.41 | 13.26 | 7.77 | 12.03 | 19.39 | 14.49 | 18.17 | 16.79 |
| | Brier | 0.38 | 0.34 | 0.39 | 0.18 | 0.42 | 0.19 | 0.16 | 0.23 | 0.21 | 0.20 | 0.11 | 0.17 | 0.28 | 0.21 | 0.27 | 0.25 |
| | NLL | 1.02 | 0.92 | 0.92 | 0.44 | 0.98 | 0.45 | 0.37 | 0.54 | 0.50 | 0.45 | 0.24 | 0.39 | 0.62 | 0.47 | 0.60 | 0.59 |

a modest fraction of extra adapter parameters, allowing us to maintain the performance boost with great parameter efficiency without any loss in original model parameters following an update.

In terms of error rate, Table 13 demonstrates that CoTTA with only `SAPIENT` adapters being adaptable has an error rate of 34.70%, surpassing CoTTA with only BN params being adaptable, which has an error rate of 32.48%.

In Fig. 6, we show the accuracy rate (%) on CIFAR100C over different percentages of added parameters in `SAPIENT`. We compare the number of trainable parameters with CoTTA as a baseline (i.e., 100% trainable parameters).

We report the number of parameters in Table 14. This illustrates the flexibility of `SAPIENT` to choose the number of parameters depending on the memory and computational budget.

To further validate `SAPIENT`'s orthogonality with source-anchoring based TTA approaches, we integrate our adapters with SANTA (Chakrabarty et al., 2023), a recent method that employs source-anchoring based self-distillation and source-guided contrastive alignment. Table 15 shows results on CIFAR10-to-CIFAR10C. `SAPIENT` adapters improve SANTA's performance from 16.21% to 15.92% mean error rate. Notably, both the groupwise and pointwise convolution components contribute to this improvement, as removing GWC yields 15.99% error. This demonstrates that `SAPIENT`'s parameter-efficient adapters can enhance various methods while maintaining the plug-and-play property.

## H   Architecture Details along with SAPIENT Adapters

In this section, we report the architecture-specific details used for adapter parameters. For CIFAR100-to-CIFAR100C experiments, we modify the widely used ResNeXt-29 architecture taken from RobustBench (Croce et al., 2021) and added adapters in between referred to as ConvAdapt layers. For ImageNet-to-ImageNetC, we modify the ResNet-50 architecture and add adapter layers in between. The added adapter layers are kept in bold.

ResNeXt-29 with adapters for CIFAR100-to-CIFAR100C

```
Hendrycks2020AugMixResNeXtNetAdpt(
        (conv_1_3x3): Conv2d(3, 64, kernel_size=(3, 3), stride=(1, 1), padding=(1, 1), bias=False)
```

Table 12: The table shows all the metrics results obtained for the CIFAR100-to-CIFAR100C online continual test-time adaptation task for the highest corruption of severity level 5 corresponding to the results obtained for `SAPIENT` reported in Table 3. `SAPIENT` here only uses 6.8% adapter parameters compared to the base model. Note that `SAPIENT` here, uses the learning objective and TTA scheme proposed by CoTTA.

| Time | | t → | | | | | | | | | | | | | | | |
|---|---|---|---|---|---|---|---|---|---|---|---|---|---|---|---|---|---|
| Method | Metric | Gaussian | shot | impulse | defocus | glass | motion | zoom | snow | frost | fog | brightness | contrast | elastic | pixelate | jpeg | Mean |
| Source | Error % | 73.00 | 68.01 | 39.37 | 29.32 | 54.11 | 30.81 | 28.76 | 39.49 | 45.81 | 50.30 | 29.53 | 55.10 | 37.23 | 74.69 | 41.25 | 46.45 |
|  | Brier | 1.11 | 1.04 | 0.58 | 0.41 | 0.79 | 0.43 | 0.40 | 0.53 | 0.64 | 0.71 | 0.41 | 0.75 | 0.51 | 1.12 | 0.56 | 0.67 |
|  | NLL | 5.59 | 4.89 | 2.00 | 1.19 | 2.86 | 1.26 | 1.16 | 1.63 | 2.12 | 2.34 | 1.16 | 2.52 | 1.50 | 5.39 | 1.74 | 2.49 |
| BN | Error % | 42.14 | 40.66 | 42.73 | 27.64 | 41.82 | 29.72 | 27.87 | 34.88 | 35.03 | 41.50 | 26.52 | 30.31 | 35.66 | 32.94 | 41.16 | 35.37 |
|  | Brier | 0.55 | 0.54 | 0.56 | 0.37 | 0.55 | 0.40 | 0.38 | 0.47 | 0.46 | 0.55 | 0.36 | 0.40 | 0.48 | 0.44 | 0.54 | 0.47 |
|  | NLL | 1.69 | 1.62 | 1.71 | 1.06 | 1.64 | 1.13 | 1.06 | 1.38 | 1.37 | 1.66 | 1.01 | 1.17 | 1.40 | 1.29 | 1.66 | 1.39 |
| TENT | Error % | 37.16 | 35.61 | 41.82 | 37.54 | 51.19 | 48.48 | 49.15 | 58.83 | 62.85 | 71.65 | 70.76 | 82.91 | 88.00 | 91.14 | 94.63 | 61.45 |
|  | Brier | 0.51 | 0.52 | 0.63 | 0.60 | 0.82 | 0.82 | 0.8585 | 1.03 | 1.13 | 1.31 | 1.32 | 1.60 | 1.68 | 1.77 | 1.85 | 1.10 |
|  | NLL | 1.49 | 1.58 | 2.14 | 2.12 | 3.28 | 3.66 | 4.17 | 5.46 | 6.71 | 8.53 | 9.04 | 14.43 | 14.17 | 16.21 | 17.66 | 7.37 |
| CoTTA | Error % | 40.09 | 37.67 | 39.77 | 26.91 | 37.82 | 28.04 | 26.26 | 32.93 | 31.72 | 40.48 | 24.72 | 26.98 | 32.33 | 28.08 | 33.46 | 32.48 |
|  | Brier | 0.53 | 0.51 | 0.53 | 0.37 | 0.50 | 0.38 | 0.36 | 0.44 | 0.43 | 0.53 | 0.35 | 0.37 | 0.44 | 0.39 | 0.45 | 0.44 |
|  | NLL | 1.60 | 1.50 | 1.58 | 1.04 | 1.47 | 1.09 | 1.02 | 1.29 | 1.24 | 1.59 | 0.96 | 1.05 | 1.25 | 1.10 | 1.30 | 1.27 |
| SAPIENT | Error % | 40.10 | 36.66 | 38.81 | 26.68 | 38.10 | 28.56 | 25.95 | 33.81 | 32.42 | 42.12 | 24.98 | 27.32 | 34.31 | 28.60 | 35.40 | 32.92 |
|  | Brier | 0.53 | 0.5 | 0.52 | 0.37 | 0.51 | 0.39 | 0.36 | 0.46 | 0.44 | 0.55 | 0.35 | 0.38 | 0.46 | 0.39 | 0.47 | 0.45 |
|  | NLL | 1.6 | 1.46 | 1.51 | 1 | 1.46 | 1.07 | 0.97 | 1.3 | 1.22 | 1.65 | 0.93 | 1.03 | 1.28 | 1.07 | 1.34 | 1.26 |

Table 13: The table compares `SAPIENT` applied over TENT and CoTTA, highlighting the orthogonality of the proposed generic framework. Results for the CIFAR100-to-CIFAR100C benchmark depict the parameter efficiency obtained (**93.2% fewer parameters**) with a meager performance drop (0.71% for TENT (100% params), 0.52% for EATA (100% params), and 0.44% for CoTTA).

| Time | t → | | | | | | | | | | | | | | | |
|---|---|---|---|---|---|---|---|---|---|---|---|---|---|---|---|---|
| Method | Gaussian | shot | impulse | defocus | glass | motion | zoom | snow | frost | fog | brightness | contrast | elastic | pixelate | jpeg | Mean |
| TENT-continual | **37.20** | 35.80 | 41.70 | 37.90 | 51.20 | 48.30 | 48.50 | 58.40 | 63.70 | 71.10 | 70.40 | 82.30 | 88.00 | 88.50 | 90.40 | 60.90 |
| TENT-continual (100% Params) | 40.27 | **35.68** | **37.28** | 26.27 | 37.81 | 28.98 | 26.97 | **33.39** | 32.52 | **39.21** | 27.33 | 32.39 | 34.87 | 32.03 | **39.65** | **33.64** |
| + SAPIENT (6.8% Params) | 41.67 | 39.40 | 41.35 | 26.96 | 40.11 | **28.87** | **26.91** | 33.87 | 33.80 | 39.94 | **26.27** | **29.55** | **34.50** | **32.02** | 40.10 | 34.35 |
| EATA-continual | 41.83 | 40.27 | 42.56 | 27.56 | 41.54 | 29.54 | 27.70 | 34.69 | 34.71 | 41.24 | 26.42 | 30.2 | 35.58 | 32.73 | 40.95 | 35.17 |
| EATA-continual (100% Params) | 41.50 | **38.81** | **41.07** | 26.82 | 39.90 | 28.90 | 26.83 | **33.56** | 33.11 | **39.60** | 25.38 | 29.00 | 34.15 | 30.79 | **38.96** | **33.89** |
| + SAPIENT (6.8% Params) | **41.21** | 38.96 | 41.24 | 26.97 | 41.07 | 29.26 | 27.13 | 33.84 | 34.3 | 39.94 | 26.04 | 30.14 | 34.61 | 31.67 | 39.74 | 34.41 |
| CoTTA (BN Params) | 40.33 | 38.3 | 40.16 | 27.67 | 39.99 | 29.76 | 27.88 | 35.51 | 34.68 | 43.4 | 26.58 | 30.52 | 35.98 | 32.05 | 37.71 | 34.70 |
| CoTTA (100% Params) | **40.09** | 37.67 | 39.77 | 26.91 | **37.82** | **28.04** | 26.26 | **32.93** | **31.72** | 40.48 | **24.72** | **26.98** | 32.33 | 28.08 | 33.46 | **32.48** |
| + SAPIENT (6.8% Params) | 40.10 | **36.66** | **38.81** | **26.68** | 38.10 | 28.56 | **25.95** | 33.81 | 32.42 | 42.12 | 24.98 | 27.32 | 34.31 | 28.60 | 35.40 | 32.92 |

Table 14: Error rate (%) on CIFAR100C over different percentages of added parameters in `SAPIENT`. Param. % in the bracket in the first two columns indicates a comparison with the base model, for e.g., 49M (7.16% of 6.90M) and 7.37M (106.80% of 6.9M). We observe that adding a similar number of trainable parameters as adapters (101.29%, last row) improves the performance over CoTTA by a small margin, and even with a much smaller number of trainable params., `SAPIENT` achieves comparable performance.

| Method | Train. Params. | Total Params. | Train. % | Error |
|---|---|---|---|---|
| **CoTTA** | 6.90M | 6.90M | 100.00% | 32.5 |
| **SAPIENT** | 0.49M (7.16%) | 7.37M (106.80%) | **6.71%** | 32.92 |
|  | 2.35M (34.12%) | 9.23M (133.76%) | 25.51% | 32.79 |
|  | 3.94M (57.14%) | 10.82M (156.78% ) | 36.45% | 32.65 |
|  | 6.99M (101.29%) | 13.89M (201.29%) | 50.32% | **32.31** |

Table 15: The table compares SAPIENT applied over SANTA. Results for the CIFAR10-to-CIFAR10C benchmark.

| Time | $t$ | | | | | | | | | | | | | | | |
|---|---|---|---|---|---|---|---|---|---|---|---|---|---|---|---|---|
| Method | Gaussian | shot | impulse | defocus | glass | motion | zoom | snow | frost | fog | brightness | contrast | elastic | pixelate | jpeg | Mean |
| SANTA | 24.01 | 19.27 | 27.22 | 11.62 | 28.01 | 12.49 | 09.64 | 14.11 | 13.60 | 12.18 | 07.74 | 10.64 | 19.07 | 14.09 | 19.51 | 16.21 |
| SANTA+SAPIENT | 23.74 | 19.59 | 27.15 | 11.72 | 27.37 | 12.21 | 10.07 | 13.93 | 13.12 | 11.94 | 07.58 | 10.14 | 17.87 | 13.56 | 18.76 | **15.92** |
| SANTA+SAPIENT (no GWC) | 23.81 | 19.57 | 27.71 | 11.36 | 27.19 | 12.42 | 09.97 | 14.07 | 13.43 | 11.95 | 07.47 | 10.62 | 18.02 | 13.36 | 18.89 | 15.99 |

```
(bn_1): BatchNorm2d(64, eps=1e-05, momentum=0.1, affine=True, track_running_stats=False)
(stage_1): Sequential(
  (0): ResNeXtBottleneck(
    (conv_reduce): Conv2d(64, 128, kernel_size=(1, 1), stride=(1, 1), bias=False)
    (bn_reduce): BatchNorm2d(128, eps=1e-05, momentum=0.1, affine=True, track_running_stats=False)
   (conv_conv): Conv2d(128, 128, kernel_size=(3, 3), stride=(1, 1), padding=(1, 1), groups=4, bias=False)
    (bn): BatchNorm2d(128, eps=1e-05, momentum=0.1, affine=True, track_running_stats=False)
    (conv_expand): Conv2d(128, 256, kernel_size=(1, 1), stride=(1, 1), bias=False)
    (bn_expand): BatchNorm2d(256, eps=1e-05, momentum=0.1, affine=True, track_running_stats=False)
    (downsample): Sequential(
      (0): Conv2d(64, 256, kernel_size=(1, 1), stride=(1, 1), bias=False)
      (1): BatchNorm2d(256, eps=1e-05, momentum=0.1, affine=True, track_running_stats=False)
    )
  )
  (1): ResNeXtBottleneck(
    (conv_reduce): Conv2d(256, 128, kernel_size=(1, 1), stride=(1, 1), bias=False)
    (bn_reduce): BatchNorm2d(128, eps=1e-05, momentum=0.1, affine=True, track_running_stats=False)
   (conv_conv): Conv2d(128, 128, kernel_size=(3, 3), stride=(1, 1), padding=(1, 1), groups=4, bias=False)
    (bn): BatchNorm2d(128, eps=1e-05, momentum=0.1, affine=True, track_running_stats=False)
    (conv_expand): Conv2d(128, 256, kernel_size=(1, 1), stride=(1, 1), bias=False)
    (bn_expand): BatchNorm2d(256, eps=1e-05, momentum=0.1, affine=True, track_running_stats=False)
  )
  (2): ResNeXtBottleneck(
    (conv_reduce): Conv2d(256, 128, kernel_size=(1, 1), stride=(1, 1), bias=False)
    (bn_reduce): BatchNorm2d(128, eps=1e-05, momentum=0.1, affine=True, track_running_stats=False)
   (conv_conv): Conv2d(128, 128, kernel_size=(3, 3), stride=(1, 1), padding=(1, 1), groups=4, bias=False)
    (bn): BatchNorm2d(128, eps=1e-05, momentum=0.1, affine=True, track_running_stats=False)
    (conv_expand): Conv2d(128, 256, kernel_size=(1, 1), stride=(1, 1), bias=False)
    (bn_expand): BatchNorm2d(256, eps=1e-05, momentum=0.1, affine=True, track_running_stats=False)
  )
)
(stage_2): Sequential(
  (0): ResNeXtBottleneck(
    (conv_reduce): Conv2d(256, 256, kernel_size=(1, 1), stride=(1, 1), bias=False)
    (bn_reduce): BatchNorm2d(256, eps=1e-05, momentum=0.1, affine=True, track_running_stats=False)
   (conv_conv): Conv2d(256, 256, kernel_size=(3, 3), stride=(2, 2), padding=(1, 1), groups=4, bias=False)
    (bn): BatchNorm2d(256, eps=1e-05, momentum=0.1, affine=True, track_running_stats=False)
    (conv_expand): Conv2d(256, 512, kernel_size=(1, 1), stride=(1, 1), bias=False)
    (bn_expand): BatchNorm2d(512, eps=1e-05, momentum=0.1, affine=True, track_running_stats=False)
    (downsample): Sequential(
      (0): Conv2d(256, 512, kernel_size=(1, 1), stride=(2, 2), bias=False)
      (1): BatchNorm2d(512, eps=1e-05, momentum=0.1, affine=True, track_running_stats=False)
    )
  )
  (1): ResNeXtBottleneckAdpt(
    (conv_reduce): Conv2d(512, 256, kernel_size=(1, 1), stride=(1, 1), bias=False)
    (bn_reduce): BatchNorm2d(256, eps=1e-05, momentum=0.1, affine=True, track_running_stats=False)
    (lhc1): ConvAdapt(
```

```
            (gwc): Conv2d(256, 256, kernel_size=(3, 3), stride=(1, 1), padding=(1, 1), groups=32)
            (pwc): Conv2d(256, 256, kernel_size=(1, 1), stride=(1, 1))
          )
         (conv_conv): Conv2d(256, 256, kernel_size=(3, 3), stride=(1, 1), padding=(1, 1), groups=4, bias=False)
          (bn): BatchNorm2d(256, eps=1e-05, momentum=0.1, affine=True, track_running_stats=False)
          (lhc2): ConvAdapt(
            (gwc): Conv2d(256, 256, kernel_size=(3, 3), stride=(1, 1), padding=(1, 1), groups=32)
            (pwc): Conv2d(256, 256, kernel_size=(1, 1), stride=(1, 1))
          )
          (conv_expand): Conv2d(256, 512, kernel_size=(1, 1), stride=(1, 1), bias=False)
          (bn_expand): BatchNorm2d(512, eps=1e-05, momentum=0.1, affine=True, track_running_stats=False)
          (lhc3): ConvAdapt(
            (gwc): Conv2d(512, 512, kernel_size=(3, 3), stride=(1, 1), padding=(1, 1), groups=64)
            (pwc): Conv2d(512, 512, kernel_size=(1, 1), stride=(1, 1))
          )
        )
        (2): ResNeXtBottleneck(
          (conv_reduce): Conv2d(512, 256, kernel_size=(1, 1), stride=(1, 1), bias=False)
          (bn_reduce): BatchNorm2d(256, eps=1e-05, momentum=0.1, affine=True, track_running_stats=False)
         (conv_conv): Conv2d(256, 256, kernel_size=(3, 3), stride=(1, 1), padding=(1, 1), groups=4, bias=False)
          (bn): BatchNorm2d(256, eps=1e-05, momentum=0.1, affine=True, track_running_stats=False)
          (conv_expand): Conv2d(256, 512, kernel_size=(1, 1), stride=(1, 1), bias=False)
          (bn_expand): BatchNorm2d(512, eps=1e-05, momentum=0.1, affine=True, track_running_stats=False)
        )
      )
      (stage_3): Sequential(
        (0): ResNeXtBottleneck(
          (conv_reduce): Conv2d(512, 512, kernel_size=(1, 1), stride=(1, 1), bias=False)
          (bn_reduce): BatchNorm2d(512, eps=1e-05, momentum=0.1, affine=True, track_running_stats=False)
         (conv_conv): Conv2d(512, 512, kernel_size=(3, 3), stride=(2, 2), padding=(1, 1), groups=4, bias=False)
          (bn): BatchNorm2d(512, eps=1e-05, momentum=0.1, affine=True, track_running_stats=False)
          (conv_expand): Conv2d(512, 1024, kernel_size=(1, 1), stride=(1, 1), bias=False)
          (bn_expand): BatchNorm2d(1024, eps=1e-05, momentum=0.1, affine=True, track_running_stats=False)
          (downsample): Sequential(
            (0): Conv2d(512, 1024, kernel_size=(1, 1), stride=(2, 2), bias=False)
            (1): BatchNorm2d(1024, eps=1e-05, momentum=0.1, affine=True, track_running_stats=False)
          )
        )
        (1): ResNeXtBottleneck(
          (conv_reduce): Conv2d(1024, 512, kernel_size=(1, 1), stride=(1, 1), bias=False)
          (bn_reduce): BatchNorm2d(512, eps=1e-05, momentum=0.1, affine=True, track_running_stats=False)
         (conv_conv): Conv2d(512, 512, kernel_size=(3, 3), stride=(1, 1), padding=(1, 1), groups=4, bias=False)
          (bn): BatchNorm2d(512, eps=1e-05, momentum=0.1, affine=True, track_running_stats=False)
          (conv_expand): Conv2d(512, 1024, kernel_size=(1, 1), stride=(1, 1), bias=False)
          (bn_expand): BatchNorm2d(1024, eps=1e-05, momentum=0.1, affine=True, track_running_stats=False)
        )
        (2): ResNeXtBottleneck(
          (conv_reduce): Conv2d(1024, 512, kernel_size=(1, 1), stride=(1, 1), bias=False)
          (bn_reduce): BatchNorm2d(512, eps=1e-05, momentum=0.1, affine=True, track_running_stats=False)
         (conv_conv): Conv2d(512, 512, kernel_size=(3, 3), stride=(1, 1), padding=(1, 1), groups=4, bias=False)
          (bn): BatchNorm2d(512, eps=1e-05, momentum=0.1, affine=True, track_running_stats=False)
          (conv_expand): Conv2d(512, 1024, kernel_size=(1, 1), stride=(1, 1), bias=False)
          (bn_expand): BatchNorm2d(1024, eps=1e-05, momentum=0.1, affine=True, track_running_stats=False)
        )
      )
      (avgpool): AdaptiveAvgPool2d(output_size=(1, 1))
      (classifier): Linear(in_features=1024, out_features=100, bias=True)
)
```

ResNet-50 with adapters for ImageNet-to-ImageNetC

```
ResNetAdapt(
    (conv1): Conv2d(3, 64, kernel_size=(7, 7), stride=(2, 2), padding=(3, 3), bias=False)
    (bn1): BatchNorm2d(64, eps=1e-05, momentum=0.1, affine=True, track_running_stats=False)
    (relu): ReLU(inplace=True)
    (maxpool): MaxPool2d(kernel_size=3, stride=2, padding=1, dilation=1, ceil_mode=False)
    (layer1): Sequential(
      (0): BottleneckAdpt(
        (conv1): Conv2d(64, 64, kernel_size=(1, 1), stride=(1, 1), bias=False)
        (lhc1): ConvAdapt(
          (gwc): Conv2d(64, 64, kernel_size=(3, 3), stride=(1, 1), padding=(1, 1), groups=8)
          (pwc): Conv2d(64, 64, kernel_size=(1, 1), stride=(1, 1))
        )
        (bn1): BatchNorm2d(64, eps=1e-05, momentum=0.1, affine=True, track_running_stats=False)
        (conv2): Conv2d(64, 64, kernel_size=(3, 3), stride=(1, 1), padding=(1, 1), bias=False)
        (lhc2): ConvAdapt(
          (gwc): Conv2d(64, 64, kernel_size=(3, 3), stride=(1, 1), padding=(1, 1), groups=8)
          (pwc): Conv2d(64, 64, kernel_size=(1, 1), stride=(1, 1))
        )
        (bn2): BatchNorm2d(64, eps=1e-05, momentum=0.1, affine=True, track_running_stats=False)
        (conv3): Conv2d(64, 256, kernel_size=(1, 1), stride=(1, 1), bias=False)
        (bn3): BatchNorm2d(256, eps=1e-05, momentum=0.1, affine=True, track_running_stats=False)
        (relu): ReLU(inplace=True)
        (downsample): Sequential(
          (0): Conv2d(64, 256, kernel_size=(1, 1), stride=(1, 1), bias=False)
          (1): BatchNorm2d(256, eps=1e-05, momentum=0.1, affine=True, track_running_stats=False)
        )
      )
      (1): BottleneckAdpt(
        (conv1): Conv2d(256, 64, kernel_size=(1, 1), stride=(1, 1), bias=False)
        (lhc1): ConvAdapt(
          (gwc): Conv2d(64, 64, kernel_size=(3, 3), stride=(1, 1), padding=(1, 1), groups=8)
          (pwc): Conv2d(64, 64, kernel_size=(1, 1), stride=(1, 1))
        )
        (bn1): BatchNorm2d(64, eps=1e-05, momentum=0.1, affine=True, track_running_stats=False)
        (conv2): Conv2d(64, 64, kernel_size=(3, 3), stride=(1, 1), padding=(1, 1), bias=False)
        (lhc2): ConvAdapt(
          (gwc): Conv2d(64, 64, kernel_size=(3, 3), stride=(1, 1), padding=(1, 1), groups=8)
          (pwc): Conv2d(64, 64, kernel_size=(1, 1), stride=(1, 1))
        )
        (bn2): BatchNorm2d(64, eps=1e-05, momentum=0.1, affine=True, track_running_stats=False)
        (conv3): Conv2d(64, 256, kernel_size=(1, 1), stride=(1, 1), bias=False)
        (bn3): BatchNorm2d(256, eps=1e-05, momentum=0.1, affine=True, track_running_stats=False)
        (relu): ReLU(inplace=True)
      )
      (2): BottleneckAdpt(
        (conv1): Conv2d(256, 64, kernel_size=(1, 1), stride=(1, 1), bias=False)
        (lhc1): ConvAdapt(
          (gwc): Conv2d(64, 64, kernel_size=(3, 3), stride=(1, 1), padding=(1, 1), groups=8)
          (pwc): Conv2d(64, 64, kernel_size=(1, 1), stride=(1, 1))
        )
        (bn1): BatchNorm2d(64, eps=1e-05, momentum=0.1, affine=True, track_running_stats=False)
        (conv2): Conv2d(64, 64, kernel_size=(3, 3), stride=(1, 1), padding=(1, 1), bias=False)
        (lhc2): ConvAdapt(
          (gwc): Conv2d(64, 64, kernel_size=(3, 3), stride=(1, 1), padding=(1, 1), groups=8)
          (pwc): Conv2d(64, 64, kernel_size=(1, 1), stride=(1, 1))
        )
        (bn2): BatchNorm2d(64, eps=1e-05, momentum=0.1, affine=True, track_running_stats=False)
```

```
      (conv3): Conv2d(64, 256, kernel_size=(1, 1), stride=(1, 1), bias=False)
      (bn3): BatchNorm2d(256, eps=1e-05, momentum=0.1, affine=True, track_running_stats=False)
      (relu): ReLU(inplace=True)
    )
  )
  (layer2): Sequential(
    (0): BottleneckAdpt(
      (conv1): Conv2d(256, 128, kernel_size=(1, 1), stride=(1, 1), bias=False)
      (lhc1): ConvAdapt(
        (gwc): Conv2d(128, 128, kernel_size=(3, 3), stride=(1, 1), padding=(1, 1), groups=16)
        (pwc): Conv2d(128, 128, kernel_size=(1, 1), stride=(1, 1))
      )
      (bn1): BatchNorm2d(128, eps=1e-05, momentum=0.1, affine=True, track_running_stats=False)
      (conv2): Conv2d(128, 128, kernel_size=(3, 3), stride=(2, 2), padding=(1, 1), bias=False)
      (lhc2): ConvAdapt(
        (gwc): Conv2d(128, 128, kernel_size=(3, 3), stride=(1, 1), padding=(1, 1), groups=16)
        (pwc): Conv2d(128, 128, kernel_size=(1, 1), stride=(1, 1))
      )
      (bn2): BatchNorm2d(128, eps=1e-05, momentum=0.1, affine=True, track_running_stats=False)
      (conv3): Conv2d(128, 512, kernel_size=(1, 1), stride=(1, 1), bias=False)
      (bn3): BatchNorm2d(512, eps=1e-05, momentum=0.1, affine=True, track_running_stats=False)
      (relu): ReLU(inplace=True)
      (downsample): Sequential(
        (0): Conv2d(256, 512, kernel_size=(1, 1), stride=(2, 2), bias=False)
        (1): BatchNorm2d(512, eps=1e-05, momentum=0.1, affine=True, track_running_stats=False)
      )
    )
    (1): BottleneckAdpt(
      (conv1): Conv2d(512, 128, kernel_size=(1, 1), stride=(1, 1), bias=False)
      (lhc1): ConvAdapt(
        (gwc): Conv2d(128, 128, kernel_size=(3, 3), stride=(1, 1), padding=(1, 1), groups=16)
        (pwc): Conv2d(128, 128, kernel_size=(1, 1), stride=(1, 1))
      )
      (bn1): BatchNorm2d(128, eps=1e-05, momentum=0.1, affine=True, track_running_stats=False)
      (conv2): Conv2d(128, 128, kernel_size=(3, 3), stride=(1, 1), padding=(1, 1), bias=False)
      (lhc2): ConvAdapt(
        (gwc): Conv2d(128, 128, kernel_size=(3, 3), stride=(1, 1), padding=(1, 1), groups=16)
        (pwc): Conv2d(128, 128, kernel_size=(1, 1), stride=(1, 1))
      )
      (bn2): BatchNorm2d(128, eps=1e-05, momentum=0.1, affine=True, track_running_stats=False)
      (conv3): Conv2d(128, 512, kernel_size=(1, 1), stride=(1, 1), bias=False)
      (bn3): BatchNorm2d(512, eps=1e-05, momentum=0.1, affine=True, track_running_stats=False)
      (relu): ReLU(inplace=True)
    )
    (2): BottleneckAdpt(
      (conv1): Conv2d(512, 128, kernel_size=(1, 1), stride=(1, 1), bias=False)
      (lhc1): ConvAdapt(
        (gwc): Conv2d(128, 128, kernel_size=(3, 3), stride=(1, 1), padding=(1, 1), groups=16)
        (pwc): Conv2d(128, 128, kernel_size=(1, 1), stride=(1, 1))
      )
      (bn1): BatchNorm2d(128, eps=1e-05, momentum=0.1, affine=True, track_running_stats=False)
      (conv2): Conv2d(128, 128, kernel_size=(3, 3), stride=(1, 1), padding=(1, 1), bias=False)
      (lhc2): ConvAdapt(
        (gwc): Conv2d(128, 128, kernel_size=(3, 3), stride=(1, 1), padding=(1, 1), groups=16)
        (pwc): Conv2d(128, 128, kernel_size=(1, 1), stride=(1, 1))
      )
      (bn2): BatchNorm2d(128, eps=1e-05, momentum=0.1, affine=True, track_running_stats=False)
      (conv3): Conv2d(128, 512, kernel_size=(1, 1), stride=(1, 1), bias=False)
      (bn3): BatchNorm2d(512, eps=1e-05, momentum=0.1, affine=True, track_running_stats=False)
```

```
        (relu): ReLU(inplace=True)
      )
      (3): BottleneckAdpt(
        (conv1): Conv2d(512, 128, kernel_size=(1, 1), stride=(1, 1), bias=False)
        (lhc1): ConvAdapt(
          (gwc): Conv2d(128, 128, kernel_size=(3, 3), stride=(1, 1), padding=(1, 1), groups=16)
          (pwc): Conv2d(128, 128, kernel_size=(1, 1), stride=(1, 1))
        )
        (bn1): BatchNorm2d(128, eps=1e-05, momentum=0.1, affine=True, track_running_stats=False)
        (conv2): Conv2d(128, 128, kernel_size=(3, 3), stride=(1, 1), padding=(1, 1), bias=False)
        (lhc2): ConvAdapt(
          (gwc): Conv2d(128, 128, kernel_size=(3, 3), stride=(1, 1), padding=(1, 1), groups=16)
          (pwc): Conv2d(128, 128, kernel_size=(1, 1), stride=(1, 1))
        )
        (bn2): BatchNorm2d(128, eps=1e-05, momentum=0.1, affine=True, track_running_stats=False)
        (conv3): Conv2d(128, 512, kernel_size=(1, 1), stride=(1, 1), bias=False)
        (bn3): BatchNorm2d(512, eps=1e-05, momentum=0.1, affine=True, track_running_stats=False)
        (relu): ReLU(inplace=True)
      )
    )
    (layer3): Sequential(
      (0): BottleneckAdpt(
        (conv1): Conv2d(512, 256, kernel_size=(1, 1), stride=(1, 1), bias=False)
        (lhc1): ConvAdapt(
          (gwc): Conv2d(256, 256, kernel_size=(3, 3), stride=(1, 1), padding=(1, 1), groups=32)
          (pwc): Conv2d(256, 256, kernel_size=(1, 1), stride=(1, 1))
        )
        (bn1): BatchNorm2d(256, eps=1e-05, momentum=0.1, affine=True, track_running_stats=False)
        (conv2): Conv2d(256, 256, kernel_size=(3, 3), stride=(2, 2), padding=(1, 1), bias=False)
        (lhc2): ConvAdapt(
          (gwc): Conv2d(256, 256, kernel_size=(3, 3), stride=(1, 1), padding=(1, 1), groups=32)
          (pwc): Conv2d(256, 256, kernel_size=(1, 1), stride=(1, 1))
        )
        (bn2): BatchNorm2d(256, eps=1e-05, momentum=0.1, affine=True, track_running_stats=False)
        (conv3): Conv2d(256, 1024, kernel_size=(1, 1), stride=(1, 1), bias=False)
        (bn3): BatchNorm2d(1024, eps=1e-05, momentum=0.1, affine=True, track_running_stats=False)
        (relu): ReLU(inplace=True)
        (downsample): Sequential(
          (0): Conv2d(512, 1024, kernel_size=(1, 1), stride=(2, 2), bias=False)
          (1): BatchNorm2d(1024, eps=1e-05, momentum=0.1, affine=True, track_running_stats=False)
        )
      )
      (1): BottleneckAdpt(
        (conv1): Conv2d(1024, 256, kernel_size=(1, 1), stride=(1, 1), bias=False)
        (lhc1): ConvAdapt(
          (gwc): Conv2d(256, 256, kernel_size=(3, 3), stride=(1, 1), padding=(1, 1), groups=32)
          (pwc): Conv2d(256, 256, kernel_size=(1, 1), stride=(1, 1))
        )
        (bn1): BatchNorm2d(256, eps=1e-05, momentum=0.1, affine=True, track_running_stats=False)
        (conv2): Conv2d(256, 256, kernel_size=(3, 3), stride=(1, 1), padding=(1, 1), bias=False)
        (lhc2): ConvAdapt(
          (gwc): Conv2d(256, 256, kernel_size=(3, 3), stride=(1, 1), padding=(1, 1), groups=32)
          (pwc): Conv2d(256, 256, kernel_size=(1, 1), stride=(1, 1))
        )
        (bn2): BatchNorm2d(256, eps=1e-05, momentum=0.1, affine=True, track_running_stats=False)
        (conv3): Conv2d(256, 1024, kernel_size=(1, 1), stride=(1, 1), bias=False)
        (bn3): BatchNorm2d(1024, eps=1e-05, momentum=0.1, affine=True, track_running_stats=False)
        (relu): ReLU(inplace=True)
      )
```

```
(2): BottleneckAdpt(
  (conv1): Conv2d(1024, 256, kernel_size=(1, 1), stride=(1, 1), bias=False)
  (lhc1): ConvAdapt(
    (gwc): Conv2d(256, 256, kernel_size=(3, 3), stride=(1, 1), padding=(1, 1), groups=32)
    (pwc): Conv2d(256, 256, kernel_size=(1, 1), stride=(1, 1))
  )
  (bn1): BatchNorm2d(256, eps=1e-05, momentum=0.1, affine=True, track_running_stats=False)
  (conv2): Conv2d(256, 256, kernel_size=(3, 3), stride=(1, 1), padding=(1, 1), bias=False)
  (lhc2): ConvAdapt(
    (gwc): Conv2d(256, 256, kernel_size=(3, 3), stride=(1, 1), padding=(1, 1), groups=32)
    (pwc): Conv2d(256, 256, kernel_size=(1, 1), stride=(1, 1))
  )
  (bn2): BatchNorm2d(256, eps=1e-05, momentum=0.1, affine=True, track_running_stats=False)
  (conv3): Conv2d(256, 1024, kernel_size=(1, 1), stride=(1, 1), bias=False)
  (bn3): BatchNorm2d(1024, eps=1e-05, momentum=0.1, affine=True, track_running_stats=False)
  (relu): ReLU(inplace=True)
)
(3): BottleneckAdpt(
  (conv1): Conv2d(1024, 256, kernel_size=(1, 1), stride=(1, 1), bias=False)
  (lhc1): ConvAdapt(
    (gwc): Conv2d(256, 256, kernel_size=(3, 3), stride=(1, 1), padding=(1, 1), groups=32)
    (pwc): Conv2d(256, 256, kernel_size=(1, 1), stride=(1, 1))
  )
  (bn1): BatchNorm2d(256, eps=1e-05, momentum=0.1, affine=True, track_running_stats=False)
  (conv2): Conv2d(256, 256, kernel_size=(3, 3), stride=(1, 1), padding=(1, 1), bias=False)
  (lhc2): ConvAdapt(
    (gwc): Conv2d(256, 256, kernel_size=(3, 3), stride=(1, 1), padding=(1, 1), groups=32)
    (pwc): Conv2d(256, 256, kernel_size=(1, 1), stride=(1, 1))
  )
  (bn2): BatchNorm2d(256, eps=1e-05, momentum=0.1, affine=True, track_running_stats=False)
  (conv3): Conv2d(256, 1024, kernel_size=(1, 1), stride=(1, 1), bias=False)
  (bn3): BatchNorm2d(1024, eps=1e-05, momentum=0.1, affine=True, track_running_stats=False)
  (relu): ReLU(inplace=True)
)
(4): BottleneckAdpt(
  (conv1): Conv2d(1024, 256, kernel_size=(1, 1), stride=(1, 1), bias=False)
  (lhc1): ConvAdapt(
    (gwc): Conv2d(256, 256, kernel_size=(3, 3), stride=(1, 1), padding=(1, 1), groups=32)
    (pwc): Conv2d(256, 256, kernel_size=(1, 1), stride=(1, 1))
  )
  (bn1): BatchNorm2d(256, eps=1e-05, momentum=0.1, affine=True, track_running_stats=False)
  (conv2): Conv2d(256, 256, kernel_size=(3, 3), stride=(1, 1), padding=(1, 1), bias=False)
  (lhc2): ConvAdapt(
    (gwc): Conv2d(256, 256, kernel_size=(3, 3), stride=(1, 1), padding=(1, 1), groups=32)
    (pwc): Conv2d(256, 256, kernel_size=(1, 1), stride=(1, 1))
  )
  (bn2): BatchNorm2d(256, eps=1e-05, momentum=0.1, affine=True, track_running_stats=False)
  (conv3): Conv2d(256, 1024, kernel_size=(1, 1), stride=(1, 1), bias=False)
  (bn3): BatchNorm2d(1024, eps=1e-05, momentum=0.1, affine=True, track_running_stats=False)
  (relu): ReLU(inplace=True)
)
(5): BottleneckAdpt(
  (conv1): Conv2d(1024, 256, kernel_size=(1, 1), stride=(1, 1), bias=False)
  (lhc1): ConvAdapt(
    (gwc): Conv2d(256, 256, kernel_size=(3, 3), stride=(1, 1), padding=(1, 1), groups=32)
    (pwc): Conv2d(256, 256, kernel_size=(1, 1), stride=(1, 1))
  )
  (bn1): BatchNorm2d(256, eps=1e-05, momentum=0.1, affine=True, track_running_stats=False)
  (conv2): Conv2d(256, 256, kernel_size=(3, 3), stride=(1, 1), padding=(1, 1), bias=False)
```

```
      (lhc2): ConvAdapt(
        (gwc): Conv2d(256, 256, kernel_size=(3, 3), stride=(1, 1), padding=(1, 1), groups=32)
        (pwc): Conv2d(256, 256, kernel_size=(1, 1), stride=(1, 1))
      )
      (bn2): BatchNorm2d(256, eps=1e-05, momentum=0.1, affine=True, track_running_stats=False)
      (conv3): Conv2d(256, 1024, kernel_size=(1, 1), stride=(1, 1), bias=False)
      (bn3): BatchNorm2d(1024, eps=1e-05, momentum=0.1, affine=True, track_running_stats=False)
      (relu): ReLU(inplace=True)
    )
  )
  (layer4): Sequential(
    (0): BottleneckAdpt(
      (conv1): Conv2d(1024, 512, kernel_size=(1, 1), stride=(1, 1), bias=False)
      (lhc1): ConvAdapt(
        (gwc): Conv2d(512, 512, kernel_size=(3, 3), stride=(1, 1), padding=(1, 1), groups=64)
        (pwc): Conv2d(512, 512, kernel_size=(1, 1), stride=(1, 1))
      )
      (bn1): BatchNorm2d(512, eps=1e-05, momentum=0.1, affine=True, track_running_stats=False)
      (conv2): Conv2d(512, 512, kernel_size=(3, 3), stride=(2, 2), padding=(1, 1), bias=False)
      (lhc2): ConvAdapt(
        (gwc): Conv2d(512, 512, kernel_size=(3, 3), stride=(1, 1), padding=(1, 1), groups=64)
        (pwc): Conv2d(512, 512, kernel_size=(1, 1), stride=(1, 1))
      )
      (bn2): BatchNorm2d(512, eps=1e-05, momentum=0.1, affine=True, track_running_stats=False)
      (conv3): Conv2d(512, 2048, kernel_size=(1, 1), stride=(1, 1), bias=False)
      (bn3): BatchNorm2d(2048, eps=1e-05, momentum=0.1, affine=True, track_running_stats=False)
      (relu): ReLU(inplace=True)
      (downsample): Sequential(
        (0): Conv2d(1024, 2048, kernel_size=(1, 1), stride=(2, 2), bias=False)
        (1): BatchNorm2d(2048, eps=1e-05, momentum=0.1, affine=True, track_running_stats=False)
      )
    )
    (1): BottleneckAdpt(
      (conv1): Conv2d(2048, 512, kernel_size=(1, 1), stride=(1, 1), bias=False)
      (lhc1): ConvAdapt(
        (gwc): Conv2d(512, 512, kernel_size=(3, 3), stride=(1, 1), padding=(1, 1), groups=64)
        (pwc): Conv2d(512, 512, kernel_size=(1, 1), stride=(1, 1))
      )
      (bn1): BatchNorm2d(512, eps=1e-05, momentum=0.1, affine=True, track_running_stats=False)
      (conv2): Conv2d(512, 512, kernel_size=(3, 3), stride=(1, 1), padding=(1, 1), bias=False)
      (lhc2): ConvAdapt(
        (gwc): Conv2d(512, 512, kernel_size=(3, 3), stride=(1, 1), padding=(1, 1), groups=64)
        (pwc): Conv2d(512, 512, kernel_size=(1, 1), stride=(1, 1))
      )
      (bn2): BatchNorm2d(512, eps=1e-05, momentum=0.1, affine=True, track_running_stats=False)
      (conv3): Conv2d(512, 2048, kernel_size=(1, 1), stride=(1, 1), bias=False)
      (bn3): BatchNorm2d(2048, eps=1e-05, momentum=0.1, affine=True, track_running_stats=False)
      (relu): ReLU(inplace=True)
    )
    (2): BottleneckAdpt(
      (conv1): Conv2d(2048, 512, kernel_size=(1, 1), stride=(1, 1), bias=False)
      (lhc1): ConvAdapt(
        (gwc): Conv2d(512, 512, kernel_size=(3, 3), stride=(1, 1), padding=(1, 1), groups=64)
        (pwc): Conv2d(512, 512, kernel_size=(1, 1), stride=(1, 1))
      )
      (bn1): BatchNorm2d(512, eps=1e-05, momentum=0.1, affine=True, track_running_stats=False)
      (conv2): Conv2d(512, 512, kernel_size=(3, 3), stride=(1, 1), padding=(1, 1), bias=False)
      (lhc2): ConvAdapt(
        (gwc): Conv2d(512, 512, kernel_size=(3, 3), stride=(1, 1), padding=(1, 1), groups=64)
```

```
          (pwc): Conv2d(512, 512, kernel_size=(1, 1), stride=(1, 1))
        )
        (bn2): BatchNorm2d(512, eps=1e-05, momentum=0.1, affine=True, track_running_stats=False)
        (conv3): Conv2d(512, 2048, kernel_size=(1, 1), stride=(1, 1), bias=False)
        (bn3): BatchNorm2d(2048, eps=1e-05, momentum=0.1, affine=True, track_running_stats=False)
        (relu): ReLU(inplace=True)
      )
    )
    (avgpool): AdaptiveAvgPool2d(output_size=(1, 1))
    (fc): Linear(in_features=2048, out_features=1000, bias=True)
  )
)
```

```python
import torch
import torch.nn as nn

class ConvAdapt(nn.Module):
    def __init__(self, in_channels, out_channels, p):
        super(ConvAdapt, self).__init__()

        # Groupwise Convolution
        self.gwc = nn.Conv2d(
            in_channels,
            out_channels,
            kernel_size=3,
            padding=1,
            groups=int(p / gp),
            bias=True,
        )
        assert in_channels == out_channels

        # Pointwise Convolution
        self.pwc = nn.Conv2d(
            in_channels, out_channels, kernel_size=1, groups=1, bias=True
        )

    def forward(self, x):
        return self.gwc(x) + self.pwc(x)
```

```python
class BasicBlockAdpt(nn.Module):
    expansion = 1

    def __init__(self, in_planes, out_planes, stride, dropRate=0.0):
        super(BasicBlockAdpt, self).__init__()
        self.bn1 = nn.BatchNorm2d(in_planes)
        self.relu1 = nn.ReLU(inplace=True)
        self.conv1 = nn.Conv2d(in_planes, out_planes, kernel_size=3, stride=stride, padding=1,
    bias=False)
        self.lhc1 = ConvAdapt(out_planes, out_planes, out_planes)  # adapter
        self.bn2 = nn.BatchNorm2d(out_planes)
        self.relu2 = nn.ReLU(inplace=True)
        self.conv2 = nn.Conv2d(out_planes, out_planes, kernel_size=3, stride=1, padding=1, bias=
    False)
        self.lhc2 = ConvAdapt(out_planes, out_planes, out_planes)  # adapter
        self.droprate = dropRate
        self.equalInOut = in_planes == out_planes
        self.convShortcut = ((not self.equalInOut) and nn.Conv2d(in_planes, out_planes,
    kernel_size=1, stride=stride, padding=0, bias=False) or None)

    def forward(self, x):
        if not self.equalInOut:
            x = self.relu1(self.bn1(x))
        else:
            out = self.relu1(self.bn1(x))
        out = self.relu2(self.bn2(self.lhc1(self.conv1(out if self.equalInOut else x))))
        if self.droprate > 0:
            out = F.dropout(out, p=self.droprate, training=self.training)
        out = self.lhc2(self.conv2(out))
        return torch.add(x if self.equalInOut else self.convShortcut(x), out)
```

