# OpenReview forum: "SAPIENT: Continual Test-time Adaptation via Lightweight plug-and-play Adapters"
_TMLR — Rejected by TMLR_

### Review · Reviewer_Hoxa · 2026-02-28

**Summary Of Contributions:**

**Summary**

This paper introduces SAPIENT, a parameter-efficient framework for continual test-time adaptation (CTTA). The core idea is to insert lightweight adapter modules (based on groupwise and pointwise convolutions) into a frozen pretrained model and update only these adapters at test time using unsupervised TTA objectives.

The proposed adapters can be seamlessly integrated into existing continual TTA pipelines (e.g., CoTTA, TENT, EATA), reducing trainable parameters by roughly 90% while maintaining comparable performance.

In addition, by everaging parameter efficiency, the authors propose a SWAG-diagonal-based approximation to maintain a posterior distribution over adapter weights during continual adaptation, enabling uncertainty estimation and improving robustness.

Finally, empirical results show comparable or improved accuracy relative to CoTTA, TENT, and related baselines, with substantial parameter and memory savings.

**Strengths**

1. Continual TTA with parameter efficiency is highly relevant for deployment in dynamic environments.

2. The plug-and-play nature makes SAPIENT easy to combine with existing methods.

3. Clear parameter-efficiency gains supported by memory and time benchmarks.

**Weaknesses**

1. There is no formal analysis of stability, forgetting bounds, or adaptation dynamics.

2. Some improvements over CoTTA are modest, particularly on classification benchmarks.

3. SAPIENT’s primary novelty lies in combining these ideas within continual TTA, rather than introducing a new adaptation principle. This makes the contribution appear more engineering-oriented than conceptually transformative.

**Audience:**

Yes

**Audience Explanation:**

TMLR’s audience includes researchers in robustness, domain adaptation, continual learning, and efficient deep learning — all of whom would likely find this work relevant.

**Claims And Evidence:**

Yes

**Claims Explanation:**

1. Parameter efficiency is clearly quantified (6–14% trainable parameters vs. 100% in CoTTA).

2. Memory and time comparisons are included.

3. Uncertainty improvements are demonstrated using NLL and Brier metrics.

**Requested Changes:**

1. Provide a sharper discussion distinguishing SAPIENT from existing methods, including EcoTTA, BeCoTTA, LoRA, and other parameter-efficient fine-tuning.

---

> ### Author Response · Authors · 2026-03-11
> **Response to Reviewer Comments**
>
> We are thankful to the reviewer for the precise evaluation of our work and the positive remarks about its relevance, runtime efficiency, and plug-and-play nature.
>
> ### Absence of Formal Stability / Forgetting Bounds
> While we acknowledge the absence of formal bounds, establishing theoretical forgetting bounds for online, unsupervised, non-stationary adaptation remains an open challenge in contemporary TTA research (unaddressed by existing TTA works such as CoTTA or TENT). However, we provide substantial **empirical evidence of stability**: SAPIENT consistently improves Round 1 performance with repeated test domains (Fig. 2 and Table 1) and sustains performance after 15 successive corruptions (Table 4) without catastrophic forgetting.
>
> ### Modest Improvements & Engineering-Oriented
> Although our classification accuracy improvements over CoTTA are occasionally minimal (for instance, Table 7: 32.92% mean error compared to CoTTA's 32.48%), achieving **comparable performance** with about **90% fewer trainable parameters** is crucial for ongoing deployment. SAPIENT (**6.8% parameters**) requires drastically less memory and significantly reduces the time for TTA.
>
> Furthermore, the application of PEFT to **unsupervised**, **source-free online** contexts requires distinct conceptual design decisions. Moreover, initializing the source-free adapters using our **identity-equivalent** methodology to mitigate representation drift and incorporating online Bayesian inference (SAPIENT-B) to enhance calibration metrics (Table 5) constitute significant research achievements that extend beyond mere engineering.
>
> ### Sharper Distinction from EcoTTA, BeCoTTA, LoRA, and PEFT
>
> Thank you for the suggestion. In the revised manuscript, we have explicitly compared SAPIENT to these methods. The key distinctions are summarized below:
>
>
> | Method                               | Source-Free | Uncertainty (UQ) | Plug-and-Play | Continual TTA |
> | ------------------------------------ | ----------------- | ---------------- | ------------- | ------------- |
> | LoRA (Hu et al., 2021)​     | ✗​       | ✗​      | ✗​   | ✗​   |
> | EcoTTA (Song et al., 2023)​ | ✗​       | ✗​      | ✗​   | ✓​   |
> | BeCoTTA (Lee et al., 2024)​ | ✗​       | ✗​      | ✗​   | ✓​   |
> | **SAPIENT (Ours)​**             | ✓​       | ✓​      | ✓​   | ✓​   |
>
> SAPIENT is the only adapter-based continual TTA method that is simultaneously **source-free**, **plug-and-play** with any unsupervised TTA loss, and capable of **Bayesian uncertainty quantification (UQ)**. We have added this comparison table and an expanded discussion to Appendix A of the revised paper and referred to it in Section 4.
>
> If you have any further questions, we will be happy to answer them.

---

### Review · Reviewer_Lmz7 · 2026-03-03

**Summary Of Contributions:**

The paper introduces SAPIENT, a highly parameter-efficient framework for continual test-time adaptation that utilizes lightweight, plug-and-play adapters with an identity-equivalent initialization to eliminate the reliance on source domain data. Its key strengths lie in its ability to seamlessly integrate with existing TTA loss functions, significantly reduce memory costs by updating roughly 10% or less of the model's parameters, and enable Bayesian uncertainty estimation for robust predictions. A minor weakness is that the approach requires making architecture-specific structural modifications to strategically insert the adapter modules between the frozen layers of a given base model.

### Strengths:
1. Clear Presentation: The paper is logically structured and clearly written, making the underlying concepts and methodology of the proposed SAPIENT framework easy to follow.

2. Simple and Effective Modules: The proposed plug-and-play adapter modules are simple in design yet highly effective, achieving comparable or better performance than state-of-the-art methods while requiring substantially fewer trainable parameters.

### Weakness:
1. Missing Important Baselines: Although the paper claims to achieve state-of-the-art performance, the experimental evaluation lacks comparisons with several important and recent baselines in the continual test-time adaptation literature.

2. Lack of Surprising Insights: Many of the paper's key findings, such as the observation that freezing the base model and updating only a small subset of parameters improves efficiency and mitigates catastrophic forgetting, are somewhat common sense and expected within the broader parameter-efficient fine-tuning literature.

**Audience:**

Yes

**Audience Explanation:**

The proposed SAPIENT framework is highly modular and easy to apply as a plug-and-play adapter to existing models without needing source domain data for initialization. Additionally, it achieves a significant optimization effect by substantially reducing the number of trainable parameters while delivering comparable or improved test-time adaptation performance.

**Broader Impact Concerns:**

No ethical problem found in this paper.

**Claims And Evidence:**

No

**Claims Explanation:**

Although the submission claims to compare SAPIENT with existing state-of-the-art (SOTA) continual test-time adaptation methods and achieve better performance, this assertion is not fully supported by the experiments. In reality, the evaluation clearly lacks comparisons with important recent baselines [1,2,3], as well as other relevant works. Without benchmarking against these critical models, the evidence for the SOTA performance claims remains unconvincing.

### References:

[1] Döbler M, Marsden R A, Yang B. Robust mean teacher for continual and gradual test-time adaptation[C]//Proceedings of the IEEE/CVF Conference on Computer Vision and Pattern Recognition. 2023: 7704-7714.

[2] Liu J, Yang S, Jia P, et al. Vida: Homeostatic visual domain adapter for continual test time adaptation[J]. arXiv preprint arXiv:2306.04344, 2023.

[3] Liu J, Xu R, Yang S, et al. Continual-mae: Adaptive distribution masked autoencoders for continual test-time adaptation[C]//Proceedings of the IEEE/CVF Conference on Computer Vision and Pattern Recognition. 2024: 28653-28663.

**Requested Changes:**

To properly substantiate the claim that the proposed method achieves comparable or better performance compared to existing SOTA continual TTA methods, it is critical to include a comprehensive comparison with more recent baselines. Including these updated comparisons is critical to securing a recommendation for acceptance, as it will accurately and fairly position the paper's contributions within the current landscape of the field.

---

> ### Author Response · Authors · 2026-03-11
> **Response to Reviewer Comments**
>
> We are thankful to Reviewer Lmz7 for positive remarks regarding the clarity of presentation and the simplicity and efficacy of the proposed plug-and-play adapter design.
>
> ### Recent Baselines Are Absent ([1], [2], [3])
> We agree that comparing these recent works would be helpful for future works; however, we would also like to emphasize that the primary contribution of this work comes from the **parameter efficiency**, i.e., with minimal parameter updates, we achieve similar performance.
> We would also like to point out the following aspects of these recent baselines compared to our approach:
> - **RMT ([1]):** Modifies the *full model*, which contradicts our objective of optimizing parameter efficiency. It is possible to concurrently employ SAPIENT and a mean-teacher objective without encountering any issues, making it an orthogonal/complementary approach to ours.
> - **ViDA ([2]):** Proposes a visual domain adapter for the CTTA problem. However, best performing results utilize **data from the source domain** to configure the adapter, which is a dependency that our **identity-equivalent initialization effectively eliminates**.
> - **Continual-MAE ([3]):** Employs a generative masked autoencoder and utilizes the Distribution-aware Masking (DaM) mechanism for better target domain knowledge extraction for the CTTA problem. However, unlike SAPIENT, their focus is **not** on making the adaptation parameter efficient.
>
> While SAPIENT may be slightly less optimal in certain contexts compared to these approaches, it remains highly competitive overall. Overall, after comparison, we observe that SAPIENT achieves competitive performance with far **fewer** trainable parameters than the existing methods. We have incorporated these baselines into the new experimental tables (highlighted in blue in the updated version of the paper).
>
> ### Lack of Surprising Insights
> We agree that Supervised PEFT is a well-studied concept. However, SAPIENT is innovative due to its distinctive and non-trivial contribution to **unsupervised, source-free continual TTA**:
> 1. **Truly Source Data Free:** Our identity-equivalent initialization enables the addition of adapters without the necessity of warm-up using the source data, hence preserving the integrity of the source knowledge during initialization.
> 2. **Practical Uncertainty Assessment:** The parameter efficiency enables real-time Bayesian inference on adapter weights through a SWAG-diagonal approximation, a feat unattainable with full-network updates.
> As noted in the common response, in accordance with standard practice, we have employed a more measured terminology and characterized our approach as a **competitive, parameter-efficient alternative** universally in the revised version of the paper.
>
> If you have any further questions, we will be happy to answer them.
>
> ### References
> [1] Döbler et al. *Robust Mean Teacher for Continual and Gradual Test-Time Adaptation*. CVPR 2023.
>
> [2] Liu et al. *ViDA: Homeostatic visual domain adapter for continual test time adaptation*. ICLR 2024.
>
> [3] Liu et al. *Continual-MAE: Adaptive Distribution Masked Autoencoders for Continual Test-Time Adaptation*. CVPR 2024.

---

### Review · Reviewer_8ufQ · 2026-03-05

**Summary Of Contributions:**

The paper introduces a parameter-efficient method designed to tackle the problem of continual test time adaptation. The method works by applying intermediate, low-parameter adapters between consecutive layers of the network giving the flexibility to decide on the overall computational budget of the method. Thanks to its parameter efficiency, the model fits into Bayesian framework giving access to parameter uncertainity estimation.

**Audience:**

Yes

**Audience Explanation:**

The topic of TTA (and in particular continual TTA) is a relatively recent topic that has gained a traction in recent years. I expect that a good part of TMLR audience might be interested in reading this piece.

**Broader Impact Concerns:**

Not applicable.

**Claims And Evidence:**

Yes

**Claims Explanation:**

The claims made by the authors are clearly supported with the empirical evaluations which are 1) sensibly chosen from the perspective of the field 2) clearly explained and 3) broad enough to expect generalizability of the results.

**Requested Changes:**

> For SAPIENT, only the adapter parameters (ω) are trainable and the rest of the network parameters (θs)
remain frozen as the source domain pre-trained weights. This disentangles the source and target domain
knowledge, preventing source domain forgetting (...).

Could you please elaborate on this, how this construction prevents from catastrophic forgetting. I understand that the only trainable part is defined by adapter parameters, but these learned parameters could in principle interfere with the original network's parameters hurting the baseline accuracy or am I missing something here?

---

> ### Author Response · Authors · 2026-03-11
> **Response to Reviewer Comments**
>
> We sincerely thank the reviewer for the **positive assessment** and the **well-posed technical question**. We are glad the reviewer found the empirical evaluations **sensibly chosen**, **clearly explained**, and **broad enough to support generalizability**.
>
> ### Q. How does freezing the base model and the specified construction prevent catastrophic forgetting?
> We agree that **freezing** the **backbone parameters** $\theta_s$ does not by itself make interference impossible, as in principle, the learned adapters $\omega$ could still perturb the forward pass and hurt the original source-model behavior.
>
> Our point is more specific. **SAPIENT reduces this risk in two ways**.
>
> First, the adapters are **initialized** to be **identity-equivalent**, so at the **start** of adaptation, the model is exactly the **same** as the **pretrained source model**, with **no initial degradation**.
>
> Second, adaptation is restricted to a **small module** rather than the full network, which **limits** where **changes** can occur and avoids overwriting the source parameters themselves.
>
> So we do **not** claim that adapter updates can never interfere in principle; rather, we claim that SAPIENT is designed to make such interference much **less likely**, and our experiments show **no** evidence of **destructive interference** in practice. In particular, across benchmarks, SAPIENT performs **comparably** or **better** than full-model adaptation baselines while using **only a small fraction** of trainable parameters, and in repeated continual adaptation settings, its performance remains stable or improves rather than degrades.
>
> We have added a **more explicit discussion** of this forgetting-prevention mechanism to **Section 3.3** of the revised paper.
>
> If you have any further questions, we will be happy to answer them.

---

### Author Response · Authors · 2026-03-11
**Common Response to All the Reviewers**

We sincerely thank all three reviewers for their thorough examination, constructive feedback, and appreciation of SAPIENT's **parameter efficiency**, **modularity**, and practical advantages. We are submitting our responses together with a revised version of the paper, with all changes highlighted in blue.

We acknowledge that using the wording of "state-of-the-art" (SOTA) for our approach when comparing it to other models may have inadvertently implied that SAPIENT regularly outperforms **all** existing continual test-time adaptation (TTA) approaches. This was **not** our intent.

In the paper (such as in Abstract and Sec. 5), we explicitly mention our primary claim that SAPIENT demonstrates **better or comparable** performance to the baselines we evaluate while using around **90% fewer trainable parameters**. We do not claim to be state-of-the-art compared to all existing or concurrent approaches; the mentioned claim introduces the **parameter efficiency** aspect, where, with minimal parameter updates, we achieve similar/better performance. This framing is in line with TMLR's evaluation criteria, which appropriately focus on whether claims are substantiated by evidence, and our claims focus on the parameter efficiency and comparable performance relative to our chosen baselines. Nevertheless, we agree that the claim mentioning the word SOTA may be misleading for the reader, and we have modified it accordingly in the updated version of the paper.

We incorporate the following in the revision:
1. Comparisons with cited recent baselines ([1], [2], [3]), with a detailed difference between the presented approach and the ones mentioned.
2. A sharper distinction between SAPIENT and existing parameter-efficient fine-tuning (PEFT) or adaptation approaches (EcoTTA, BeCoTTA, LoRA).
3. An explicit clarification of how SAPIENT prevents catastrophic forgetting, directly addressing the mechanism behind **freezing the backbone** and using **identity-equivalent initialization**.
4. Revised the phrasing throughout to align with the clarified performance claims.

We hope our answers and the revised manuscript address your concerns.
We would be happy to answer any further questions.

### References
[1] Döbler et al. *Robust Mean Teacher for Continual and Gradual Test-Time Adaptation*. CVPR 2023.

[2] Liu et al. *ViDA: Homeostatic visual domain adapter for continual test time adaptation*. ICLR 2024.

[3] Liu et al. *Continual-MAE: Adaptive Distribution Masked Autoencoders for Continual Test-Time Adaptation*. CVPR 2024.

---

### Decision · Action_Editor_y3P8 · 2026-04-07

**Recommendation:** Reject

**Audience:**

Yes

**Audience Explanation:**

The studied problem is important, with many new methods pushing the state of the art and improving our capabilities in this challenge. The new ideas provided here are valuable; if they are framed correctly, I believe readers would find them interesting.

However, given all the reasons provided in the previous point, I must recommend rejection of this paper. While certain aspects are valuable, the current version does not meet TMLR's publication standards.

Overall, I believe the method needs to be repositioned relative to existing methods (PEFT and others, see my previous comment) to help readers understand the approach's strengths and weaknesses. This would help future work by building on the method to address its limitations.

**Claims And Evidence:**

No

**Claims Explanation:**

The paper studies the important problem of test time adaptation of vision models. The problem has recently gained significant attention and is important to the community. The paper presents a practical, parameter-efficient approach to adapting vision models, but, as several reviewers noted, it has several non-negligible flaws. The reviewers acknowledged the clarity of presentation and the practicality of the method, but they also raised two concerns that remain. First, they mentioned that the main mechanism, namely freezing the backbone and adapting a small subset of parameters via lightweight modules, falls within the PEFT regime of solutions and does not include an actual comparison with PEFT methods or new insights into this specific PEFT formulation. In fact, the new way proposed here to add the adaptor seems interesting, but adds layers to the network, therefore increasing inference time and latency; none of these caveats are mentioned in the paper. I expect an ML paper to be transparent about all its limitations; this is the only way the average reader can understand whether they want to use the method.

Second, the empirical evaluation does not fully substantiate the broader claims made in the paper. As Reviewer Lmz7 noted, the experimental section omits several important recent baselines. Given the existence of stronger source-free continual test-time adaptation (CTTA) and prototype-guided methods evaluated on the same benchmarks, the reported classification results do not support the authors' claim that their approach "achieves better or similar performance compared to existing state-of-the-art continual TTA methods." Notably, methods such as SANTA (Chakrabarty et al.), Parameter-Selective CTTA (Tian et al.), and other efficient approaches that outperform the proposed method (as observed on CIFAR-10 and CIFAR-100) are missing from the comparison. While I do not expect every new method to achieve state-of-the-art results, I concur with Reviewer Lmz7 that greater transparency in the empirical evaluation is necessary.

**Resubmission Of Major Revision:**

The authors may consider submitting a major revision at a later time.